# Complex Variable Solution for Stress and Displacement of Layered Soil with Finite Thickness

**Xiangyu Sha [1], Aizhong Lu [1,*], Hui Cai [1] and Chonglin Yin [2]**

[1] Institute of Hydroelectric and Geotechnical Engineering, North China Electric Power University, Beijing 102206, China; shaxiangyu@ncepu.edu.cn (X.S.); caihui@ncepu.edu.cn (H.C.)
[2] POWERCHINA Chengdu Engineering Corporation Limited, Chengdu 610072, China; yinchonglin@ncepu.edu.cn
[*] Correspondence: lvaizhong@ncepu.edu.cn

**Abstract:** The static problem of a layered isotropic elastic body is a very useful research subject in relation to the analysis and design of foundation works. Due to the complexity of the problem, there is no analytical solution to the problem so far. This study provides an efficient analytical approach to accurately calculate the displacement and stress fields of the soil. The constraints of bedrock on soil, different soil layer thickness and the shear stress of the foundation on soil were all taken into account in the analysis. In this study, each layer is regarded as an isotropic elastomer with infinite width, and the layers are in complete contact. By using conformal mapping, each layer is mapped to a unit circle, and the two complex potential functions are expanded into Taylor series with unknown coefficients. These unknown coefficients are obtained by satisfying boundary conditions and continuity conditions. The boundary and continuity conditions were verified in this paper. As a validation step, we compared the analytical results for the settlement with the results of the ANSYS numerical simulations and found good agreement. Parametric analyses were also carried out to investigate the influence of different distribution forms of base pressure on surface settlement, and the effects of layered properties on the surface settlement and stress field.

**Keywords:** strip foundation; layered soil; bedrock; stress and displacement; complex variable solution



## 1. Introduction

In many places, it is common to see compressible soil overlying rigid bedrock. In this case, the stress and displacement of soil are obviously different from that of a semi-infinite body medium [1]. Natural soils tend to be horizontally layered during deposition, the density and hardness of each layer are generally different, and the deformation characteristics may vary greatly [2]. When the stress in the soil is too large, the deformation of the soil will cause unacceptable settlement of the building and even the instability of the entire soil. Therefore, an accurate calculation of soil stress and displacement is an important basis for the stability analysis of building foundations and geotechnical structures [3].

Finite element method can provide useful solutions to many complex geological conditions. However, extensive computational time is required, especially when complete parametric analyses need to be performed [4]. Analytical solutions provide an efficient and quick approach to gain insight into the nature of the problem [5]; in addition, they enable analyses of a wide range of parameter values so that the physics of the problem can be better understood [4]. In the past, researchers mainly used the integral transformation method to analyze the elastic static problems of layered soil. Burmister [6] provided the integral expression of stress and displacement of two layers of soil under a flexible circular foundation based on the integral transformation method, and also established an empirical formula for surface settlement when both layers of soil consist of incompressible materials. In many studies in the literature, the soil surface is subjected to a circular uniformly distributed load, which is considered a spatial axisymmetric problem. Both layers of soil

are linear elastomers and in complete contact. The second layer of soil was infinitely deep. Then, smooth contact [7] and three-layer [8] soil with complete contact were considered. Because only the integral transformation equations of stress and displacement could be derived by the integral transformation method, many numerical methods were proposed to solve the integral equations. Barovich et al. [9] studied the stress field of two-layer elastomer under elliptical distributed load, in which the second-layer elastomer was a semi-infinite body, and presented a numerical value of the stress on the axis of symmetry. Chen [10] identified the shortcomings of this method, that is, the integral converges slowly or does not even converge at the boundary. Chen [10] improved the method to solve the problem of the stress field and displacement field of three-layer soil under circular and rectangular loads. However, the subsoil was still semi-infinite, which means this method was not suitable for bedrock at the bottom of the soil. Since then, many scholars have used the integral transformation method to study the layered material problem, but the bottom layer was assumed to be a semi-infinite body [11–28].

When there is bedrock under the soil, it is very difficult to obtain an analytical solution regarding the stress and displacement due to the constraint of the bedrock on the soil. At present, only Lu et al. [1] have obtained an explicit analytical expression for the stress and displacement of single-layer finite thickness soil under the action of a strip foundation by a complex variable method. In this paper, the solution in [1] is extended to layered soil, and the complex variable method [29] is used to solve the stress and displacement of multi-layer soil above bedrock under the action of a strip shallow foundation. Under the action of a strip foundation, the soil is subjected to a strip-distributed load. It is assumed that the distributed load does not change along the direction perpendicular to the surface; thus, the problem solved in this paper is a plane strain problem [1]. In this paper, each layer of soil is regarded as an isotropic linear elastic body with finite thickness, and the layers are in complete contact with each other. Each layer of soil is mapped into a unit circle by using a conformal transformation tool in the complex variable method, and the two complex potential functions of each layer of soil can be expanded into a Taylor series. According to the stress boundary condition of the surface, the displacement boundary condition of the soil bottom, and the stress and displacement continuity conditions on the contact surface, the equation for solving the complex potential function coefficients can be obtained, and through the complex potential function, the exact solution for the stress and displacement at any point in the soil can be obtained. Differently to the integral transformation method, for any given load, this method can directly provide an explicit expression of the stress and displacement in each layer of soil. At the same time, the corresponding FORTRAN program compiled in this paper is characterized by fast calculation speed and high calculation accuracy, which are of great benefit to engineering applications.

## 2. Problem Statement and Method Presentation

This paper deals with the case of complete contact between layers of multi-layer soil and complete constraint with bedrock below it. When the soil surface is subjected to distributed load, the deformation of bedrock is considered to be negligible compared with the deformation of soil, that is, the displacement component on the interface between the soil and bedrock is equal to zero. Figure 1a shows a problem in which the upper boundary is the stress boundary condition, the lower boundary is the displacement boundary condition, and the contact surface of each layer is the stress and displacement continuity condition.

In Figure 1a, the four end points $(A, C, D, F)$ of each layer of soil are points at infinity. $t$ is any point on the contact surface. There is a vector relation about $t$:

$$\overrightarrow{O_2O_1} + \overrightarrow{O_1t} = \overrightarrow{O_2t} \tag{1}$$

As shown in Figure 1, the $j$-th layer ($j = 1,2,\cdots,n$) soil is mapped to the inner domain of the unit circle in the $\zeta_j$ plane. The mapping function is:

$$z_j = \omega_j(\zeta_j) = \frac{H_j}{\pi}\ln\left(\frac{1+\zeta_j}{1-\zeta_j}\right) \qquad (2)$$

where $z_j$ is any point in the $j$-th layer soil, $z_j = x_j + iy_j$, $i = \sqrt{-1}$; $H_j$ is the thickness of the $j$-th layer soil. $\zeta_j$ is the point where $z_j$ maps to the image plane. $\zeta_j = \rho_j e^{i\theta_j}$, $\rho_j = 1$ corresponds to the upper and lower boundaries in the $j$-th layer soil, and $\sigma_j$ is used to denote $\zeta_j$ at this time.

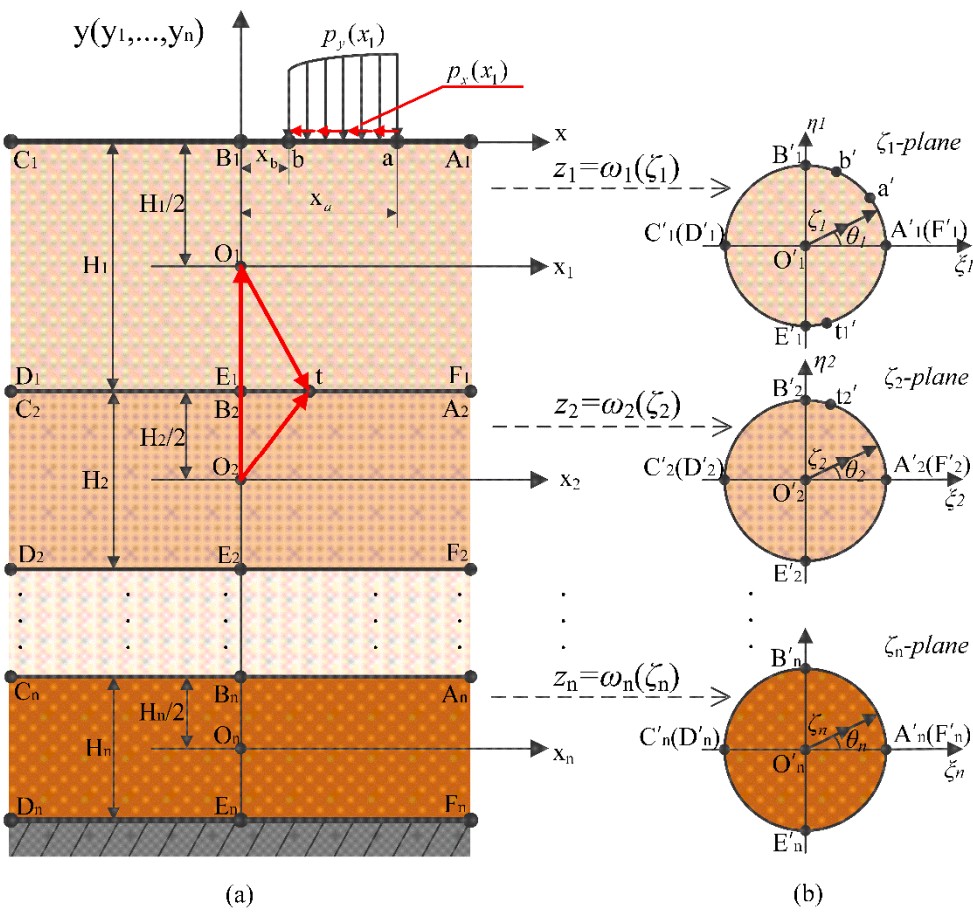

**Figure 1.** The multi-layer soil above the bedrock is mapped on to the inner domain of the unit circle in the image plane: (**a**) multi-layer soil subjected to strip load and (**b**) each layer is mapped to the inner domain of the unit circle in the image plane.

As shown in Figure 1, in the $\zeta_1$ plane, the circumference of the unit circle ($\sigma_1 = 1$) corresponds to the upper and lower boundaries of the first layer of soil. $A_1$, $B_1$, $C_1$, $D_1$, $E_1$, $F_1$ are mapped to $A_1'$, $B_1'$, $C_1'$, $D_1'$, $E_1'$, $F_1'$, respectively. The upper boundary ($\overline{A_1 B_1 C_1}$) is mapped to the upper semicircle ($\overparen{A_1' B_1' C_1'}$). The lower boundary ($\overline{D_1 E_1 F_1}$) is mapped to the lower semicircle ($\overparen{D_1' E_1' F_1'}$). The coordinate origin ($O_1$) is mapped to the center of the unit circle ($O_1'$) of the $\zeta_1$ plane. On the upper and lower boundaries, all points of $x_1 \to +\infty$ are mapped to the same point ($\sigma_1 = 1$) of the $\zeta_1$ plane, and all points of $x_1 \to -\infty$ are mapped to the same point ($\sigma_1 = -1$) of the $\zeta_1$ plane. The coordinate axes $x_1$ and $y_1$ are mapped to $\xi_1$ and $\eta_1$ axes. The points $a$ and $b$ are mapped to $a'$ and $b'$. The rest of the layers can be deduced by analogy.

By substituting Equation (2) into Equation (1), the relationship between $\theta_1$ and $\theta_2$ at the same point $t$ on the contact surface of the 1st layer and the 2nd layer can be obtained:

$$\theta_2 = \arccos\left(\frac{T_1{}^2 - 1}{T_1{}^2 + 1}\right) \tag{3}$$

where $T_1 = \left(-sin\theta_1 / (1 - cos\theta_1)\right)^{H_1/H_2}$.

In the same way, on the contact surface of the $j$-th layer and the $j + 1$-th layer:

$$\theta_{j+1} = \arccos\left(\frac{T_j{}^2 - 1}{T_j{}^2 + 1}\right) \tag{4}$$

where $T_{j+1} = \left(-sin\theta_j / (1 - cos\theta_j)\right)^{H_j/H_{j+1}}$, $j = 1, 2, \ldots, n-1$.

$\sigma_{j,x}$, $\sigma_{j,y}$ and $\tau_{j,xy}$ are the horizontal normal stress, vertical normal stress and shear stress of the $j$-th layer soil, respectively, and $u_j$ and $v_j$ are the horizontal displacement and vertical displacement of the $j$-th layer soil, respectively. The three stress components $\sigma_{j,x}$, $\sigma_{j,y}$, $\tau_{j,xy}$ and two displacement components $u_j$, $v_j$ of point $z_j$ can be expressed as [21]:

$$\begin{cases} \sigma_{j,x} + \sigma_{j,y} = 4\mathrm{Re}\left(\varphi'_{1,j}(z_j)\right) \\ \sigma_{j,z} = \mu_j\left(\sigma_{j,x} + \sigma_{j,y}\right) \end{cases} \tag{5}$$

$$\sigma_{j,y} - \sigma_{j,x} + 2i\tau_{j,xy} = 2\left[\overline{z_j}\varphi''_{1,j}(z_j) + \psi'_{1,j}(z_j)\right] \tag{6}$$

$$u_j + iv_j = \frac{1}{2G_j}\left[\kappa_j\varphi_{1,j}(z_j) - z_j\overline{\varphi'_{1,j}(z_j)} - \overline{\psi_{1,j}(z_j)}\right] \tag{7}$$

where $\kappa_j = 3 - 4\mu_j$, and $\mathrm{Re}[\,\ldots\,]$ means to take the real part of $[\,\ldots\,]$, the imaginary part represented by $\mathrm{Im}[\,\ldots\,]$ will also be involved later. $G_j$ and $\mu_j$ are the shear modulus and Poisson's ratio of the $j$-th layer soil, respectively. The single valued analytic functions of the $j$-th layer are $\varphi_{1,j}(z_j)$ and $\psi_{1,j}(z_j)$. The superscripts $(.)'$ and $(.)''$ denote the first and second derivatives of $(.)$ with respect to $z_j$.

For the problem analyzed in this paper, the stress boundary conditions can be obtained from Equations (5) and (6):

$$2\mathrm{Re}[\varphi_{1,1}(z_1)] + \overline{z_1}\varphi''_{1,1}(z_1) + \psi'_{1,1}(z_1) = \begin{cases} 0, x_1 \in (x_{1a}, \infty) \\ p(x_1) + iq(x_1), x_1 \in [x_{1b}, x_{1a}] \\ 0, x_1 \in (-\infty, x_{1b}) \end{cases} \tag{8}$$

where $z_1$ is the point on the boundary of $\overline{A_1B_1C_1}$, $p_x(x_1)$ is the horizontal load and $p_y(x_1)$ is the vertical load.

The continuity condition of displacement and stress on the contact surface between the lower boundary of the $j$-th layer and the upper boundary of the $j + 1$-th layer can be obtained from Equations (5)–(7).

$$\frac{1}{G_j}\left[\kappa_j\varphi_{1,j}(z_j) - z_j\overline{\varphi'_{1,j}(z_j)} - \overline{\psi_{1,j}(z_j)}\right] = \frac{1}{G_{j+1}}\left[\kappa_{j+1}\varphi_{1,j+1}(z_{j+1}) - z_{j+1}\overline{\varphi'_{1,j+1}(z_{j+1})} - \overline{\psi_{1,j+1}(z_{j+1})}\right] \tag{9}$$

$$2\mathrm{Re}\left[\varphi_{1,j}(z_j)\right] + \overline{z_j}\varphi''_{1,j}(z_j) + \psi'_{1,j}(z_j) = 2\mathrm{Re}\left[\varphi_{1,j+1}(z_{j+1})\right] + \overline{z_{j+1}}\varphi''_{1,j+1}(z_{j+1}) + \psi'_{1,j+1}(z_{j+1}) \tag{10}$$

In Equations (9) and (10), $j = 1, 2, \ldots, n-1$. $z_j$ and $z_{j+1}$ are the same point on the contact surface, where $z_j$ is on $\overline{D_jE_jF_j}$, and $z_{j+1}$ is on $\overline{A_jB_jC_j}$.

Assuming that the bottom boundary is completely constrained, i.e., the horizontal and vertical displacements are equal to zero, the displacement boundary condition on $\overline{D_n E_n F_n}$ can be obtained from Equation (7):

$$\kappa_n \varphi_{1,n}(z_n) - z_n \varphi'_{1,n}(z_n) - \overline{\psi_{1,n}(z_n)} = 0 \tag{11}$$

Substituting Equation (2) into $\varphi_{1,j}(z_j)$ and $\psi_{1,j}(z_j)$, we can obtain $\varphi_j(\zeta_j)$ and $\psi_j(\zeta_j)$. In the unit circle, they can be expressed as follows:

$$\begin{cases} \varphi_j(\zeta_j) = \sum\limits_{k=0}^{\infty} a_{j,k} \zeta_j^k \\ \psi_j(\zeta_j) = \sum\limits_{k=0}^{\infty} b_{j,k} \zeta_j^k \end{cases} (j = 1, 2, \ldots, n) \tag{12}$$

where $a_{j,k}$ and $b_{j,k}$ are the complex coefficients to be solved:

$$\begin{cases} a_{j,k} = a_{j,k,R} + i a_{j,k,I} \\ b_{j,k} = b_{j,k,R} + i b_{j,k,I} \end{cases} \tag{13}$$

In Equation (13), $a_{j,k,R}$ and $a_{j,k,I}$ are the real part and imaginary part of $a_{j,k}$ respectively; $b_{j,k,R}$ and $b_{j,k,I}$ are the real part and imaginary part of $b_{j,k}$ respectively. This can be solved by Equations (8)–(11).

On the boundary, $\zeta_j$ is denoted by $\sigma_j$, Equations (8)–(11) in the image plane can be rewritten as

$$2\text{Re}\left[\frac{\varphi'_1(\sigma_1)}{\omega'_1(\sigma_1)}\right] + \frac{\overline{\omega_1(\sigma_1)}}{[\omega'_1(\sigma_1)]^2}\left[\varphi''_1(\sigma_1) - \frac{\omega''_1(\sigma_1)\varphi'_1(\sigma_1)}{\omega'_1(\sigma_1)}\right] + \frac{\psi'_1(\sigma_1)}{\omega'_1(\sigma_1)} = \begin{cases} 0, \theta_1 \in [0, \theta_{1a'}) \\ p(x_1) + iq(x_1), \theta_1 \in [\theta_{1a'}, \theta_{1b'}] \\ 0, \theta_1 \in (\theta_{1b'}, \pi] \end{cases} \tag{14}$$

$$\frac{1}{G_j}\left[\kappa_j \varphi_j(\sigma_j) - \frac{\omega_j(\sigma_j)}{\omega'_j(\sigma_j)}\overline{\varphi'_j(\sigma_j)} - \overline{\psi_j(\sigma_j)}\right] = \\ \frac{1}{G_{j+1}}\left[\kappa_{j+1}\varphi_{j+1}(\sigma_{j+1}) - \frac{\omega_{j+1}(\sigma_{j+1})}{\omega'_{j+1}(\sigma_{j+1})}\overline{\varphi'_{j+1}(\sigma_{j+1})} - \overline{\psi_{j+1}(\sigma_{j+1})}\right] \tag{15}$$

$$2\text{Re}\left[\frac{\varphi'_j(\sigma_j)}{\omega'_j(\sigma_j)}\right] + \frac{\overline{\omega_j(\sigma_j)}}{[\omega'_j(\sigma_j)]^2}\left[\varphi''_j(\sigma_j) - \frac{\omega''_j(\sigma_j)\varphi'_j(\sigma_j)}{\omega'_j(\sigma_j)}\right] + \frac{\psi'_j(\sigma_j)}{\omega'_j(\sigma_j)} = \\ 2\text{Re}\left[\frac{\varphi'_{j+1}(\sigma_{j+1})}{\omega'_{j+1}(\sigma_{j+1})}\right] + \frac{\overline{\omega_{j+1}(\sigma_{j+1})}}{[\omega'_{j+1}(\sigma_{j+1})]^2}\left[\varphi''_{j+1}(\sigma_{j+1}) - \frac{\omega''_{j+1}(\sigma_{j+1})\varphi'_{j+1}(\sigma_{j+1})}{\omega'_{j+1}(\sigma_{j+1})}\right] + \frac{\psi'_{j+1}(\sigma_{j+1})}{\omega'_{j+1}(\sigma_{j+1})} \tag{16}$$

$$\kappa_n \varphi_n(\sigma_n) - \frac{\omega_n(\sigma_n)}{\omega'_n(\sigma_n)}\overline{\varphi'_n(\sigma_n)} - \overline{\psi_n(\sigma_n)} = 0 \tag{17}$$

In Equation (14)

$$x_1 = \frac{H_1}{\pi}\ln\left(\frac{\sin\theta_1}{1 - \cos\theta_1}\right) \tag{18}$$

$$\omega_1(\sigma_1) = \frac{H_1}{\pi}\ln\left(\frac{\sin\theta_1}{1 - \cos\theta_1}\right) + \frac{H_1}{2}i, \theta_1 \in [0, \pi] \tag{19}$$

In Equations (15) and (16), $j = 1, \ldots, n-1$:

$$\begin{cases} \omega(\sigma_j) = \frac{H_j}{\pi}\ln\left(\frac{-\sin\theta_j}{1-\cos\theta_j}\right) - \frac{H_j}{2}i, \theta_j \in [\pi, 2\pi] \\ \omega(\sigma_{j+1}) = \frac{H_{j+1}}{\pi}\ln\left(\frac{\sin\theta_{j+1}}{1-\cos\theta_{j+1}}\right) + \frac{H_{j+1}}{2}i, \theta_{j+1} \in [0, \pi] \end{cases} \tag{20}$$

In Equation (17)

$$\omega_n(\sigma_n) = \frac{H_n}{\pi}\ln\left(\frac{-\sin\theta_n}{1 - \cos\theta_n}\right) - \frac{H_n}{2}i, \theta_n \in [\pi, 2\pi] \tag{21}$$

In this paper, the power series method is used to solve $\varphi_j(\zeta_j)$ and $\psi_j(\zeta_j)$.

### 3. Solving Process

As shown in Figure 1b, in the image plane, we divide the surface $\overset{\frown}{A_1' B_1' C_1'}$ into three intervals $([\delta, \theta_{a'}), [\theta_{a'}, \theta_{b'}], (\theta_{b'}, \pi - \delta])$, and divide them equally into $m_{1,1}$, $m_{1,2}$ and $m_{1,3}$ parts, respectively. Take $m_1 = m_{1,1} + m_{1,2} + m_{1,3}$, $\delta$ is a small quantity. When $\theta = 0$ or $2\pi$, Equations (12)–(15) are meaningless, so we take $\delta = 0.01$. The contact surface of two adjacent layers ($\overset{\frown}{D_j' E_j' F_j'}$ and $\overset{\frown}{A_{j+1}' B_{j+1}' C_{j+1}'}$) is divided equally into $m_{j+1}$ parts $(j = 1, \ldots, n - 1)$. $\overset{\frown}{D_n' E_n' F_n'}$ is divided equally into $m_{n+1}$ parts. Then each node in the first layer has:

$$\begin{cases} \theta_{1,s} = \delta + \frac{\theta_{1a'} - \delta}{m_{1,1}}(s - 1), & s = 1, \cdots, m_{1,1} \\ \theta_{1,m_{1,1}+s} = \theta_{1a'} + \frac{\theta_{1b'} - \theta_{1a'}}{m_{1,2}}(s - 1), & s = 1, \cdots, m_{1,2} + 1 \\ \theta_{1,m_{1,1}+m_{1,2}+1+s} = \theta_{1b'} + \frac{\pi - \delta - \theta_{1b'}}{m_{1,3}}s, & s = 1, \cdots, m_{1,3} \\ \theta_{1,m_1+1+s} = \pi + \delta + \frac{\pi - 2\delta}{m_2}(s - 1), & s = 1, \cdots, m_2 + 1 \\ \sigma_{1,s} = \cos\theta_{1,s} + i\sin\theta_{1,s}, & s = 1, \cdots, m_1 + m_2 + 2 \end{cases} \quad (22)$$

where:

$$\begin{cases} \theta_{1a'} = \arccos\left( \dfrac{e^{\frac{2\pi}{H_1}x_{1a}} - 1}{e^{\frac{2\pi}{H_1}x_{1a}} + 1} \right) \\ \theta_{1b'} = \arccos\left( \dfrac{e^{\frac{2\pi}{H_1}x_{1b}} - 1}{e^{\frac{2\pi}{H_1}x_{1b}} + 1} \right) \end{cases} \quad (23)$$

Each node in the *j*-th layer has:

$$\begin{cases} \theta_{j,s} = \delta + \frac{\pi - 2\delta}{m_j}(s - 1), & s = 1, \cdots, m_j + 1 \\ \theta_{j,m_j+1+s} = \pi + \delta + \frac{\pi - 2\delta}{m_{j+1}}(s - 1), & s = 1, \cdots, m_{j+1} + 1 \\ \sigma_{j,s} = \cos\theta_{j,s} + i\sin\theta_{j,s}, & s = 1, \cdots, m_j + m_{j+1} + 2 \end{cases} \quad (j = 2, 3, \ldots, n - 1) \quad (24)$$

Each node in the *n*-th layer has

$$\begin{cases} \theta_{n,s} = \delta + \frac{\pi - 2\delta}{m_n}(s - 1), & s = 1, \cdots, m_n + 1 \\ \theta_{n,m_n+1+s} = \pi + \delta + \frac{\pi - 2\delta}{m_{n+1}}(s - 1), & s = 1, \cdots, m_{n+1} + 1 \\ \sigma_{n,s} = \cos\theta_{n,s} + i\sin\theta_{n,s}, & s = 1, \cdots, m_n + m_{n+1} + 2 \end{cases} \quad (25)$$

By substituting $\sigma_{1,s}(s = 1, \ldots, m_1 + 1)$ in Equation (22) into Equation (14), $\sigma_{1,s}(s = m_1 + 1, \ldots, m_1 + m_2 + 1)$ in Equation (22), $\sigma_{j,s}$ in Equation (24), and $\sigma_{n,s}(s = 1, \ldots, m_n + 1)$ in Equation (25) into Equations (15) and (16), and $\sigma_{n,s}(s = m_n + 2, \ldots, m_{n+1} + 1)$ in Equation (25) into Equation (17), we can obtain the infinite linear equations for $a_{j,k}$, $\overline{a_{j,k}}$, $b_{j,k}$ and $\overline{b_{j,k}}$ $(j = 1, \ldots, n)$.

This paper adopts the power series method, $\varphi_j(\zeta_j)$ and $\psi_j(\zeta_j)$ are taken as finite terms, and the highest powers are $n_{j,1}$ and $n_{j,2}$ respectively.

$$\begin{cases} \varphi_j(\zeta_j) = \sum_{k=0}^{n_{j,1}} a_{j,k}\zeta_j^k \\ \psi_j(\zeta_j) = \sum_{k=0}^{n_{j,2}} b_{j,k}\zeta_j^k \end{cases} \quad (j = 1, 2, \ldots, n) \quad (26)$$

By combining boundary conditions and continuity conditions, a system of linear equations for $a_{j,k}$, $\overline{a_{j,k}}$, $b_{j,k}$ and $\overline{b_{j,k}}$ can be obtained

$$2R_e\left(\sum_{k=1}^{n_{1,1}} c_{1,ks}a_{1,k}\right) + \sum_{k=1}^{n_{1,1}} d_{1,ks}a_{1,k} - \sum_{k=1}^{n_{1,1}} e_{1,ks}a_{1,k} + \sum_{k=1}^{n_{1,2}} f_{1,ks}b_{1,k} = f_s \tag{27}$$

$$\begin{aligned}
&\sum_{k=1}^{n_{j,1}} g_{j,ks}a_{j,k} - \sum_{k=1}^{n_{j,1}} h_{j,ks}\overline{a_{j,k}} - \sum_{k=1}^{n_{j,2}} l_{j,ks}\overline{b_{j,k}} + \frac{\kappa_j}{G_j}a_{j,0} - \frac{1}{G_j}\overline{b_{j,0}} - \\
&\sum_{k=1}^{n_{j+1,1}} g_{j+1,ks}a_{j+1,k} + \sum_{k=1}^{n_{j+1,1}} h_{j+1,ks}\overline{a_{j+1,k}} + \sum_{k=1}^{n_{j+1,2}} l_{j+1,ks}\overline{b_{j+1,k}} - \frac{\kappa_{j+1}}{G_{j+1}}a_{j+1,0} + \frac{1}{G_{j+1}}\overline{b_{j+1,0}} = 0
\end{aligned} \tag{28}$$

$$\begin{aligned}
&2\mathrm{Re}\left(\sum_{k=1}^{n_{j,1}} c_{j,ks}a_{j,k}\right) + \sum_{k=1}^{n_{j,1}} d_{j,ks}a_{j,k} - \sum_{k=1}^{n_{j,1}} e_{j,ks}a_{j,k} + \sum_{k=1}^{n_{j,2}} f_{j,ks}b_{j,k} - \\
&2\mathrm{Re}\left(\sum_{k=1}^{n_{j+1,1}} c_{j+1,ks}a_{j+1,k}\right) - \sum_{k=1}^{n_{j+1,1}} d_{j+1,ks}a_{j+1,k} + \sum_{k=1}^{n_{j+1,1}} e_{j+1,ks}a_{j+1,k} - \sum_{k=1}^{n_{j+1,2}} f_{j+1,ks}b_{j+1,k} = 0
\end{aligned} \tag{29}$$

$$\sum_{k=1}^{n_{n,1}} g_{n,ks}a_{n,k} - \sum_{k=1}^{n_{n,1}} h_{n,ks}\overline{a_{n,k}} - \sum_{k=1}^{n_{n,2}} l_{n,ks}\overline{b_{n,k}} + \frac{\kappa_n}{G_n}a_{n,0} - \frac{1}{G_n}\overline{b_{n,0}} = 0 \tag{30}$$

In Equation (27), $s = 1,\dots,m_1+1$. In Equations (28) and (29), for the $j$-th layer $s = m_j+2, m_j+3,\dots,m_j+m_{j+1}+2$, for the $j+1$-th layer $s = m_{j+1}+1, m_{j+1},\dots,1$ ($j = 1,\dots,n-1$). In Equation (30), $s = m_n+2,\dots,m_n+m_{n+1}+2$. The coefficients $c_{j,ks}$, $d_{j,ks}$, $e_{j,ks}$, $f_{j,ks}$, $g_{j,ks}$, $h_{j,ks}$, $l_{j,ks}$, $f_j$ in Equations (27)–(30) are provided in Appendix A.

Equations (31) and (32) can be obtained from the real and imaginary parts of Equation (27).

$$\begin{aligned}
&\sum_{k=1}^{n_{1,1}} \mathrm{Re}[2c_{1,ks} + d_{1,ks} - e_{1,ks}]a_{1,k,R} + \sum_{k=1}^{n_{1,1}} \mathrm{Im}[-2c_{1,ks} - d_{1,ks} + e_{1,ks}]a_{1,k,I} + \\
&\sum_{k=1}^{n_{1,2}} \mathrm{Re}[f_{1,ks}]b_{1,k,R} - \sum_{k=1}^{n_{1,2}} \mathrm{Im}[f_{1,ks}]b_{1,k,I} = \mathrm{Re}[f_s]
\end{aligned} \tag{31}$$

$$\begin{aligned}
&\sum_{k=1}^{n_{1,1}} \mathrm{Im}[d_{1,ks} - e_{1,ks}]a_{1,k,R} + \sum_{k=1}^{n_{1,1}} \mathrm{Re}[d_{1,ks} - e_{1,ks}]a_{1,k,I} + \\
&\sum_{k=1}^{n_{1,2}} \mathrm{Im}[f_{1,ks}]b_{1,k,R} + \sum_{k=1}^{n_{1,2}} \mathrm{Re}[f_{1,ks}]b_{1,k,I} = \mathrm{Im}[f_s]
\end{aligned} \tag{32}$$

Equations (33) and (34) can be obtained from the real and imaginary parts of Equation (28).

$$\begin{aligned}
&\sum_{k=1}^{n_{j,1}} \mathrm{Re}\left[g_{j,ks} - h_{j,ks}\right]a_{j,k,R} - \sum_{k=1}^{n_{j,1}} \mathrm{Im}\left[g_{j,ks} + h_{j,ks}\right]a_{j,k,I} - \\
&\sum_{k=1}^{n_{j,2}} \mathrm{Re}\left[l_{j,ks}\right]b_{j,k,R} - \sum_{k=1}^{n_{j,2}} \mathrm{Im}\left[l_{j,ks}\right]b_{j,k,R} + \frac{\kappa_j}{G_j}a_{j,0,R} - \frac{1}{G_j}b_{j,0,R} - \\
&\sum_{k=1}^{n_{j+1,1}} \mathrm{Re}\left[g_{j+1,ks} - h_{j+1,ks}\right]a_{j+1,k,R} + \sum_{k=1}^{n_{j+1,1}} \mathrm{Im}\left[g_{j+1,ks} + h_{j+1,ks}\right]a_{j+1,k,I} + \\
&\sum_{k=1}^{n_{j+1,2}} \mathrm{Re}\left[l_{j+1,ks}\right]b_{j+1,k,R} + \sum_{k=1}^{n_{j+1,2}} \mathrm{Im}\left[l_{j+1,ks}\right]b_{j+1,k,R} - \frac{\kappa_{j+1}}{G_{j+1}}a_{j,0,R} + \frac{1}{G_{j+1}}b_{j+1,0,R} = 0
\end{aligned} \tag{33}$$

$$\begin{aligned}
&\sum_{k=1}^{n_{j,1}} \mathrm{Im}\left[g_{j,ks} - h_{j,ks}\right]a_{j,k,R} + \sum_{k=1}^{n_{j,1}} \mathrm{Re}\left[g_{j,ks} + h_{j,ks}\right]a_{j,k,I} - \\
&\sum_{k=1}^{n_{j,2}} \mathrm{Im}\left[l_{j,ks}\right]b_{j,k,R} + \sum_{k=1}^{n_{j,2}} \mathrm{Re}\left[l_{j,ks}\right]b_{j,k,I} + \frac{\kappa_j}{G_j}a_{j,0,I} + \frac{1}{G_j}b_{j,0,I} - \\
&\sum_{k=1}^{n_{j+1,1}} \mathrm{Im}\left[g_{j+1,ks} - h_{j+1,ks}\right]a_{j+1,k,R} - \sum_{k=1}^{n_{j+1,1}} \mathrm{Re}\left[g_{j+1,ks} + h_{j+1,ks}\right]a_{j+1,k,I} + \\
&\sum_{k=1}^{n_{j+1,2}} \mathrm{Im}\left[l_{j+1,ks}\right]b_{j+1,k,R} - \sum_{k=1}^{n_{j+1,2}} \mathrm{Re}\left[l_{j+1,ks}\right]b_{j+1,k,I} - \frac{\kappa_{j+1}}{G_{j+1}}a_{j+1,0,I} - \frac{1}{G_{j+1}}b_{j+1,0,I} = 0
\end{aligned} \tag{34}$$

Equations (35) and (36) can be obtained from the real and imaginary parts of Equation (29).

$$
\begin{aligned}
&\sum_{k=1}^{n_{j,1}} \mathrm{Re}\Big[2c_{j,ks} + d_{j,ks} - e_{j,ks}\Big]a_{j,k,R} + \sum_{k=1}^{n_{j,1}} \mathrm{Im}\Big[-2c_{j,ks} - d_{j,ks} + e_{j,ks}\Big]a_{j,k,I} + \\
&\sum_{k=1}^{n_{j,2}} \mathrm{Re}\Big[f_{j,ks}\Big]b_{j,k,R} - \sum_{k=1}^{n_{j,2}} \mathrm{Im}\Big[f_{j,ks}\Big]b_{j,k,I} - \\
&\sum_{k=1}^{n_{j+1,1}} \mathrm{Re}\Big[2c_{j+1,ks} + d_{j+1,ks} - e_{j+1,ks}\Big]a_{j+1,k,R} - \sum_{k=1}^{n_{j+1,1}} \mathrm{Im}\Big[-2c_{j+1,ks} - d_{j+1,ks} + e_{j+1,ks}\Big]a_{j+1,k,I} - \\
&\sum_{k=1}^{n_{j+1,2}} \mathrm{Re}\Big[f_{j+1,ks}\Big]b_{j+1,k,R} + \sum_{k=1}^{n_{j+1,2}} \mathrm{Im}\Big[f_{j+1,ks}\Big]b_{j+1,k,I} = 0
\end{aligned}
\tag{35}
$$

$$
\begin{aligned}
&\sum_{k=1}^{n_{j,1}} \mathrm{Im}\Big[d_{j,ks} - e_{j,ks}\Big]a_{j,k,R} + \sum_{k=1}^{n_{j,1}} \mathrm{Re}\Big[d_{j,ks} - e_{j,ks}\Big]a_{j,k,I} + \\
&\sum_{k=1}^{n_{j,2}} \mathrm{Im}\Big[f_{j,ks}\Big]b_{j,k,R} + \sum_{k=1}^{n_{j,2}} \mathrm{Re}\Big[f_{j,ks}\Big]b_{j,k,I} - \\
&\sum_{k=1}^{n_{j+1,1}} \mathrm{Im}\Big[d_{j+1,ks} - e_{j+1,ks}\Big]a_{j+1,k,R} - \sum_{k=1}^{n_{j+1,1}} \mathrm{Re}\Big[d_{j+1,ks} - e_{j+1,ks}\Big]a_{j+1,k,I} - \\
&\sum_{k=1}^{n_{j+1,2}} \mathrm{Im}\Big[f_{j+1,ks}\Big]b_{j+1,k,R} - \sum_{k=1}^{n_{j+1,2}} \mathrm{Re}\Big[f_{j+1,ks}\Big]b_{j+1,k,I} = 0
\end{aligned}
\tag{36}
$$

Equations (37) and (38) can be obtained from the real and imaginary parts of Equation (30).

$$
\begin{aligned}
&\sum_{k=1}^{n_{n,1}} \mathrm{Re}[g_{n,ks} - h_{n,ks}]a_{n,k,R} - \sum_{k=1}^{n_{n,1}} \mathrm{Im}[g_{n,ks} + h_{n,ks}]a_{n,k,I} - \\
&\sum_{k=1}^{n_{n,2}} \mathrm{Re}[l_{n,ks}]b_{n,k,R} - \sum_{k=1}^{n_{n,2}} \mathrm{Im}[l_{n,ks}]b_{n,k,R} + \frac{\kappa_n}{G_n}a_{n,0,R} - \frac{1}{G_n}b_{n,0,R} = 0
\end{aligned}
\tag{37}
$$

$$
\begin{aligned}
&\sum_{k=1}^{n_{n,1}} \mathrm{Im}[g_{n,ks} - h_{n,ks}]a_{n,k,R} + \sum_{k=1}^{n_{n,1}} \mathrm{Re}[g_{n,ks} + h_{n,ks}]a_{n,k,I} - \\
&\sum_{k=1}^{n_{n,2}} \mathrm{Im}[l_{n,ks}]b_{n,k,R} + \sum_{k=1}^{n_{n,2}} \mathrm{Re}[l_{n,ks}]b_{n,k,I} + \frac{\kappa_n}{G_n}a_{n,0,I} + \frac{1}{G_n}b_{n,0,I} = 0
\end{aligned}
\tag{38}
$$

Since $a_{j,0}$ and $b_{j,0}$ do not affect the stress, they represent the translation of the rigid body. If $a_{j,0}$ or $b_{j,0}$ is equal to zero, it will not affect the calculation result of displacement, so in this paper, $b_{j,0} = 0$.

The system of linear equations of $a_{j,k,R}$, $a_{j,k,I}$, $b_{j,k,R}$, $b_{j,k,I}$ can be obtained from Equations (31)–(38).

$$
\boldsymbol{DX} = \boldsymbol{F}
\tag{39}
$$

where: $\boldsymbol{X} = \left( \cdots\cdots \underbrace{a_{j,1,R}\ a_{j,1,I} \cdots a_{j,n_{j,1},R}\ a_{j,n_{j,1},I}\ b_{j,1,R}\ b_{j,1,I} \cdots b_{j,n_{j,2},R}\ b_{j,n_{j,2},I}\ a_{j,0,R}\ a_{j,0,I}}_{2n_{j,1}+2n_{j,2}+2\ \text{coefficients in the } j-\text{th layer}} \cdots\cdots \right)^{\mathrm{T}}$

$(j = 1,\ldots,n)$, $\boldsymbol{F} = \left( \mathrm{Re}[f_1] \cdots \mathrm{Re}\big[f_{m_1+1}\big]\ \mathrm{Im}[f_1] \cdots \mathrm{Im}\big[f_{m_1+1}\big]0\ldots0 \right)^{\mathrm{T}}$. There are $\sum_{j=1}^{n} 2\big(m_j + m_{j+1} + 2\big)$ terms in matrix $\boldsymbol{F}$. The elements in coefficient matrix $\boldsymbol{D}$ are coefficients of Equations (29)–(36) for $a_{j,k,R}$, $a_{j,k,I}$, $b_{j,k,R}$, $b_{j,k,I}$, and the size is row $\sum_{j=1}^{n} 2\big(m_j + m_{j+1} + 2\big)$ and column $\sum_{j=1}^{n} 2\big(n_{j1} + n_{j2} + 1\big)$.

In order to ensure the accuracy of calculation, $n_j$ needs to take a large value. After calculation, it is found that the result obtained by Equation (39) may be very unsatisfactory. In order to obtain satisfactory results, this paper uses ordinary least squares. Taking $\sum_{j=1}^{n} 2\big(m_j + m_{j+1} + 2\big) > \sum_{j=1}^{n} 2\big(n_{j1} + n_{j2} + 1\big)$:

$$
\boldsymbol{X} = \left( \boldsymbol{D}^T\boldsymbol{D} \right)^{-1}\boldsymbol{D}^T\boldsymbol{F}
\tag{40}
$$

The analytic functions $\varphi_j(\zeta_j)$ and $\psi_j(\zeta_j)$ can be obtained from Equation (40).

The displacement and stress of any point in the $j$-th layer can be calculated by Equations (41) and (42), respectively.

$$2G_j(u_j + iv_j) = \kappa_j \varphi_j(\zeta_j) - \frac{\omega_j(\zeta_j)}{\overline{\omega'_j(\zeta_j)}}\overline{\varphi'_j(\zeta_j)} - \overline{\psi_j(\zeta_j)} \tag{41}$$

$$\begin{cases} \sigma_{j,y} + \sigma_{j,x} = 4\mathrm{Re}\left[\dfrac{\varphi'_j(\zeta_j)}{\omega'_j(\zeta_j)}\right] \\[2ex] \sigma_{j,y} - \sigma_{j,x} + 2i\tau_{j,xy} = 2\dfrac{\overline{\omega_j(\zeta_j)}}{[\overline{\omega'_j(\zeta_j)}]^2}\left[\varphi''_j(\zeta_j) - \dfrac{\omega''_j(\zeta_j)\varphi'_j(\zeta_j)}{\omega'_j(\sigma_j)}\right] + 2\dfrac{\psi'_j(\zeta_j)}{\omega'_j(\zeta_j)} \end{cases} \tag{42}$$

From Equations (41) and (42), we have

$$\begin{cases} u_j = \dfrac{\mathrm{Re}\left(\kappa_j t_{j,4} - \frac{t_{j,1}}{t_{j,2}}\overline{t_{j,5}} - \overline{t_{j,7}}\right)}{2G_j} \\[3ex] v_j = \dfrac{\mathrm{Im}\left(\kappa_j t_{j,4} - \frac{t_{j,1}}{t_{j,2}}\overline{t_{j,5}} - \overline{t_{j,7}}\right)}{2G_j} \end{cases} \tag{43}$$

$$\begin{cases} \sigma_{j,x} = (AA_j - BB_j)/2 \\ \sigma_{j,y} = (AA_j + BB_j)/2 \\ \tau_{j,xy} = CC_j/2 \end{cases} \tag{44}$$

where

$$\begin{aligned} AA_j &= 4\mathrm{Re}\left[t_{j,5}/t_{j,2}\right] \\ BB_j &= 2\mathrm{Re}\left[\overline{t_{j,1}}(t_{j,6} - t_{j,3}t_{j,5}/t_{j,2})/t_{j,2}^2 + t_{j,8}/t_{j,2}\right] \\ CC_j &= 2\mathrm{Im}\left[\overline{t_{j,1}}(t_{j,6} - t_{j,3}t_{j,5}/t_{j,2})/t_{j,2}^2 + t_{j,8}/t_{j,2}\right] \end{aligned} \tag{45}$$

where $t_{j,1}$ $t_{j,8}$ are functions of $\zeta_j$; the details are provided in Appendix B.

## 4. Analysis and Discussion

In order to ensure accuracy, the values of $m_j$, $n_{j,1}$, $n_{j,2}$ in this paper are large, which means matrix **D** is too large, thus we took two layers of soil as an example. The example only discusses the effect of a smooth foundation on soil, which means $p_x(x_1) = 0$.

### 4.1. Verification of Boundary Conditions and Continuity Conditions

Taking $H_1 = 30m$, $H_2 = 20m$, $\mu_1 = 0.2$, $\mu_2 = 0.3$, $x_a = 2$ m, $x_b = -2$ m, $\delta = 0.01$, $n_{1,1} = n_{1,2} = n_{2,1} = n_{2,2} = 550$, $m_{1,1} = m_{1,3} = 700$, $m_{1,2} = 80$, $m_2 = 800$, $m_3 = 600$, $G_1 = 1.0$, $G_2 = 2.0$, $p_y(x_1) = -3(4 - x_1^2)/8$, $p_x(x_1) = 0$. The unit of $p_y(x_1)$, $G_1$, $G_2$ is not given here. If the unit of $p_y(x_1)$ is $kPa$ and the unit of $G_1$, $G_2$ is MPa, the unit of stress and displacement are kPa and mm, respectively.

According to the given known conditions, the solution process in Section 3 can be used to calculate the $\varphi_j(\zeta_j)$ and $\psi_j(\zeta_j)$ of each layer ($j = 1, 2$), and then the stress components $\sigma_{1,y}$ and $\tau_{1,xy}$ of the surface can be obtained by Equation (42). If $\sigma_{1,y} = p_y(x_1)$, $\tau_{1,xy} = p_x(x_1) = 0$, the stress boundary condition is satisfied. It can be seen from the calculation that the value of $\tau_{1,xy}$ on the surface is very small, basically equal to zero, and $\sigma_{1,y}$ is also basically equal to $p_y(x_1)$. Figure 2 shows the distribution curve of $\sigma_{1,y}$ and $p_y(x_1)$ at the surface. It can be seen that the vertical stress curve of the surface basically coincides with the load curve, which indicates that the stress boundary conditions on the surface are well satisfied.

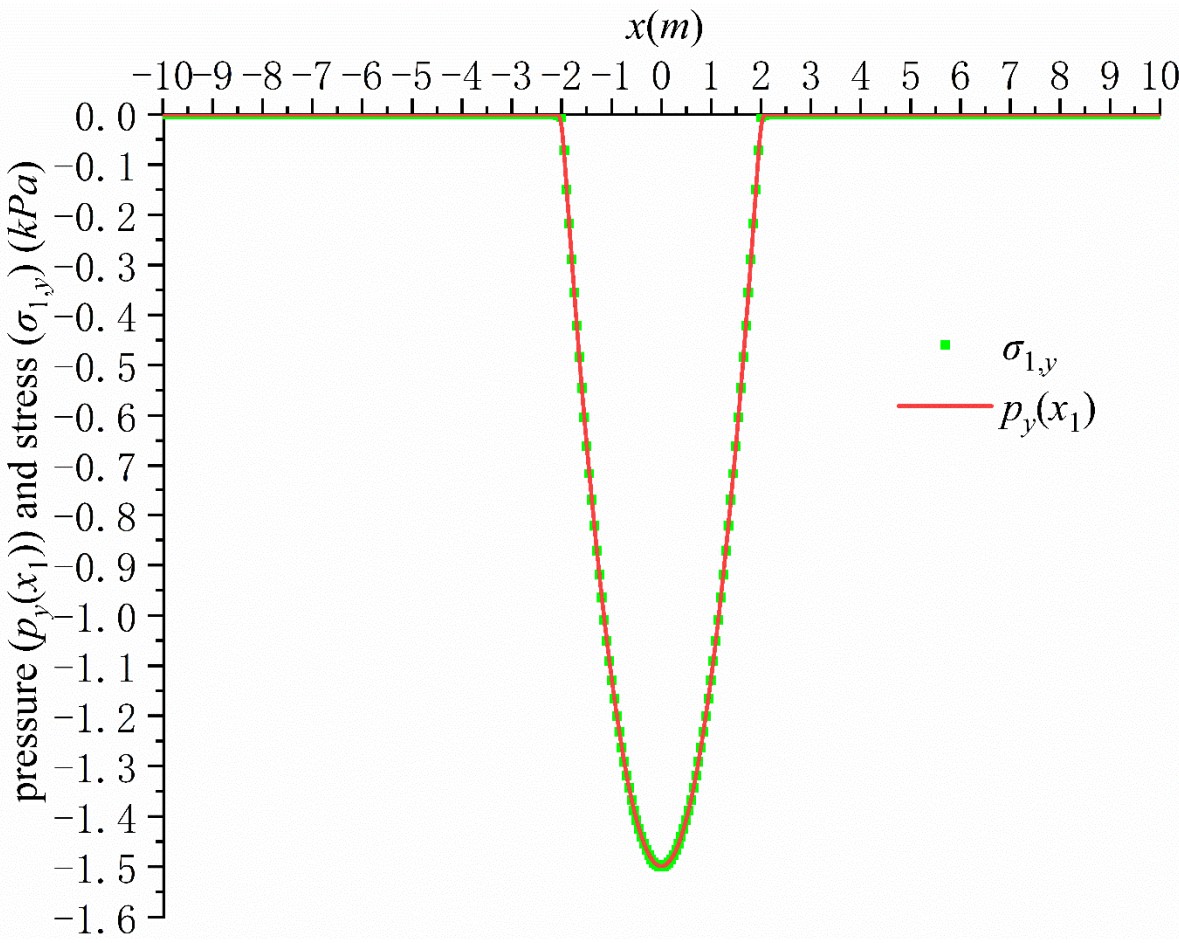

**Figure 2.** Verification of stress boundary conditions.

According to the calculated $\varphi_j(\zeta_j)$ and $\psi_j(\zeta_j)$, the displacement and stress of each layer on the contact surface can also be obtained through Equations (41) and (42), therefore the continuity condition of the displacement and stress can be verified. Figure 3 shows the verification of the displacement continuity condition. It can be seen that $u_1 \approx u_2$, $v_1 \approx v_2$, where $u_1$ and $v_1$ are the $x$-direction displacement and $y$-direction displacement of the lower boundary of the first layer of soil, respectively, while $u_2$ and $v_2$ are the $x$-direction displacement and $y$-direction displacement of the same position of the upper boundary of the second layer of soil, respectively. Figure 4 shows the verification of the stress continuity condition, which shows that $\sigma_{1,y} \approx \sigma_{2,y}$, $\tau_{1,xy} \approx \tau_{2,xy}$. It can be seen from Figures 3 and 4 that the displacement and stress continuity condition can also be satisfied very well. The maximum displacement in Figure 3 is $-0.255$ mm, and compared with this, the maximum displacement of the lower boundary of the second layer is $1.3791294 \times 10^{-4}$ mm, which means the displacement boundary conditions of the lower boundary of the second layer are also very well satisfied.



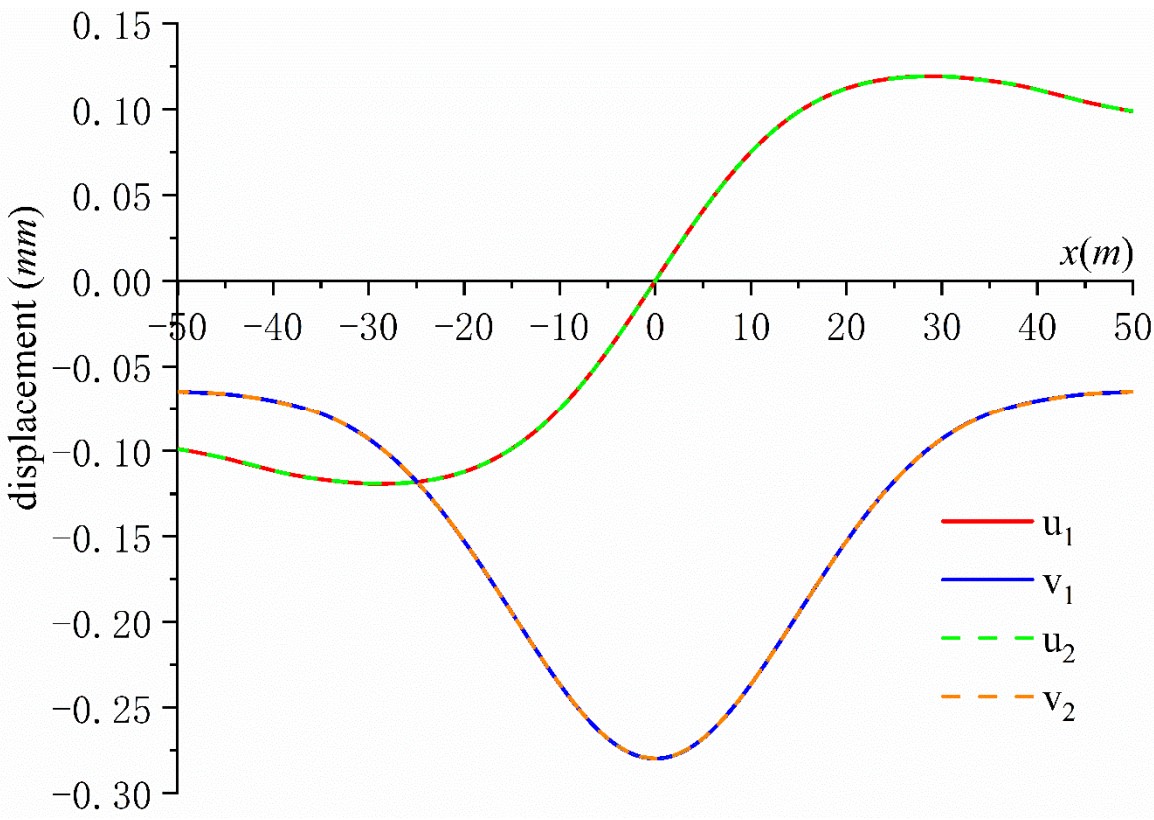

**Figure 3.** Verification of displacement continuity condition.

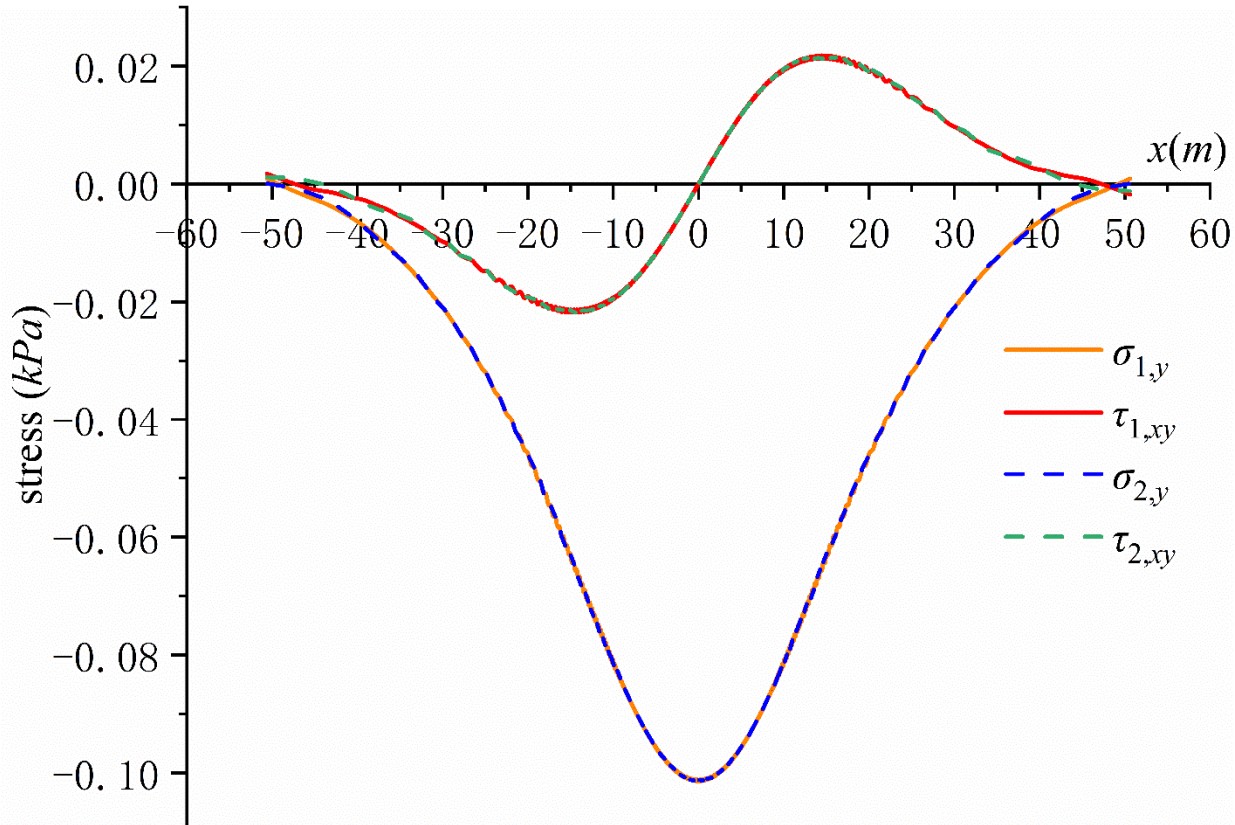

**Figure 4.** Verification of stress continuity condition.

### 4.2. Comparison with Numerical Method

In Section 4.1, the surface stress boundary conditions, the displacement boundary condition of the lower boundary of the second layer and the stress and displacement continuity conditions between layers were verified. It was found that the boundary condition and the continuity conditions were very well satisfied; however, this is not enough to establish that the whole derivation process presented in this paper is correct. In order to verify the correctness of the whole derivation process, it is necessary to compare the calculation results of this paper with those of the numerical method. When the results of the two methods are in good agreement, the whole derivation process and the calculation program used in this paper are correct.

The schematic diagram of the ANSYS [30] model is shown in Figure 5. The model is divided into upper and lower soil layers. The width of the upper soil model is 100 m, the thickness is 30 m, the load action interval is $-2m \leq x_1 \leq 2m, y = 0$, the element type is plane42, the elastic modulus is 2.4 MPa, the Poisson's ratio is 0.2, The element size in most areas is 0.5 (shown in Figure 6), and the element size is 0.05 in the area of $-2m \leq x_1 \leq 2m, -10m \leq y \leq 0$ (shown in Figure 7),. The width of the lower soil model is 100 m, the thickness is 20 m, the element type is plane42, the elastic modulus is 5.2 MPa, the Poisson's ratio is 0.3, and the element size is 0.5. The types of contact elements are targe169 (lower contact surface) and conta171 (upper contact surface). Horizontal displacement constraints are applied to the left and right sides of the model, and horizontal displacement constraints and vertical displacement constraints are applied to the bottom of the model. A parabolic distributed load is applied to the top of the model. The command flow is "*f,i,fy,(nx(i)×nx(i)−4.0) ∗ 3/160*". There are 70,849 nodes and 70,684 elements in the model. Figure 8 shows the ANSYS deformation diagram. Figure 9 shows a comparison between the analytical solution and numerical solution of the surface settlement. ANSYS version 15.0 was used in this paper.

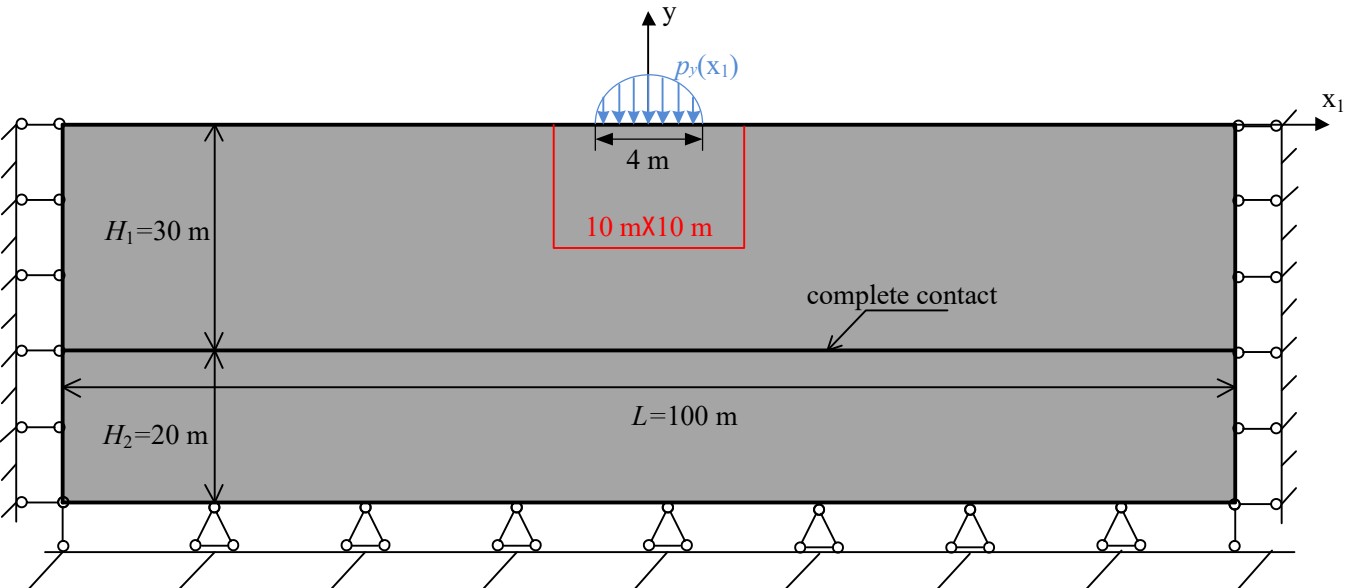

**Figure 5.** Schematic diagram of ANSYS model.

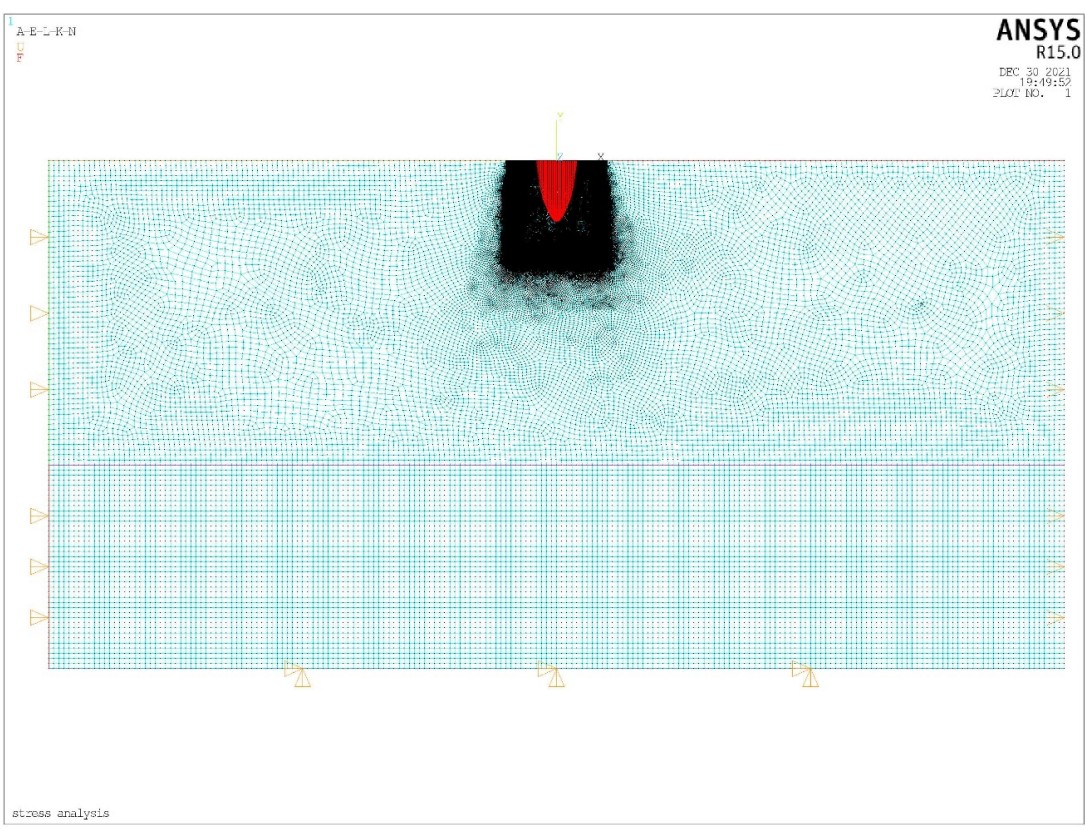

**Figure 6.** ANSYS model diagram.

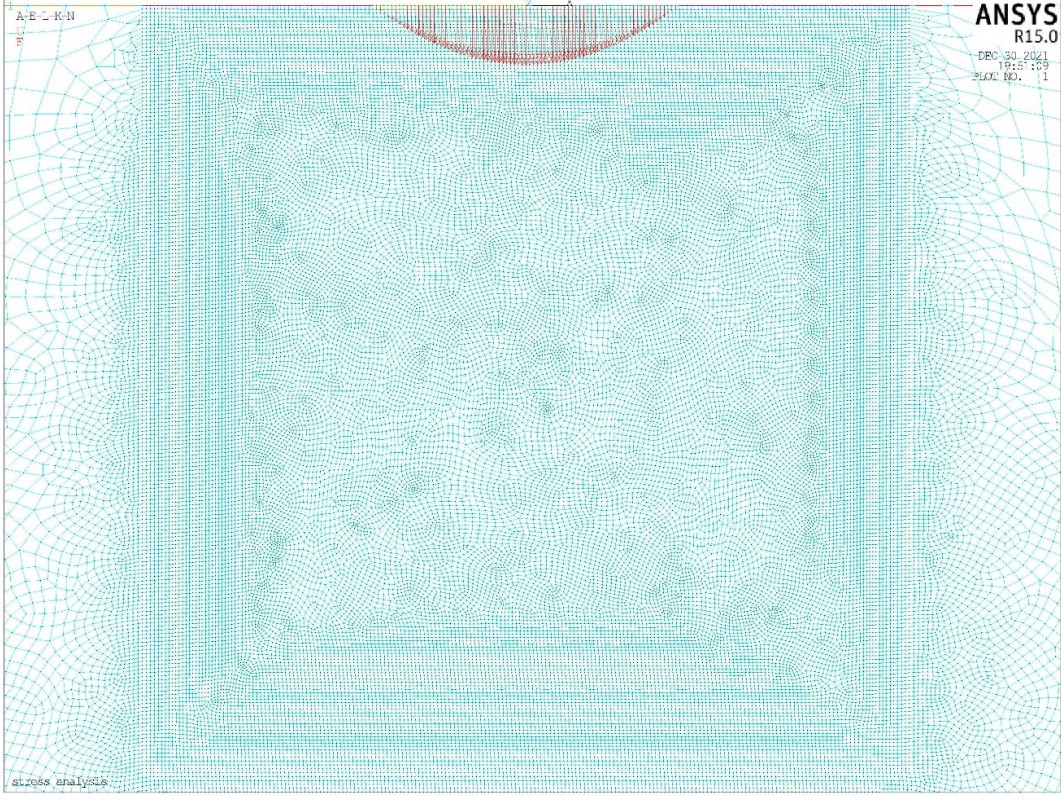

**Figure 7.** Distribution of elements near the load.

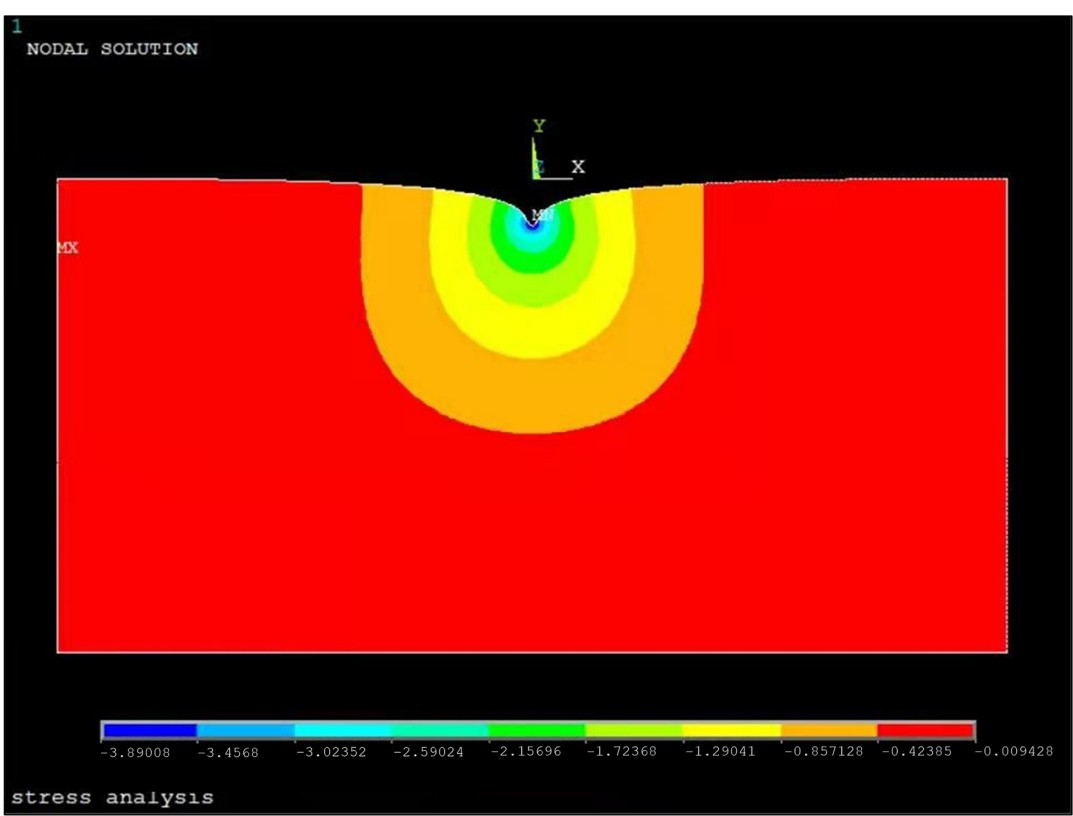

**Figure 8.** The deformation diagram of ANSYS.

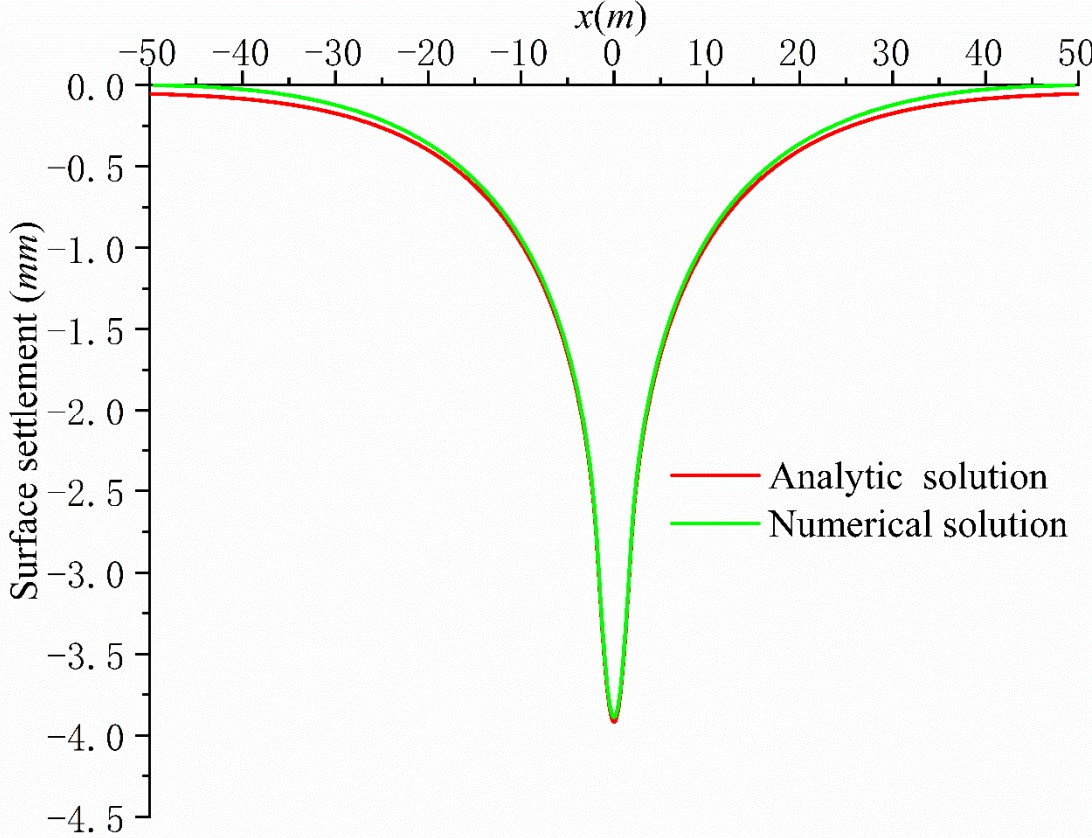

**Figure 9.** Comparison between analytical solution and numerical solution of surface settlement.

*4.3. Influence of Base Pressure Distribution on Surface Settlement*

As shown in Figure 10a, three types of base pressure distribution under vertical action are discussed:

(1)     The concentration of base pressure is large at the edge: $p_{y1}(x_1) = \left(-3x_1^2 - 12\right)/16$;

(2)     The concentration of base pressure is large at the center: $p_{y2}(x_1) = -3\left(4 - x_1^2\right)/8$;

(3)     Uniform distribution of base pressure: $p_{y3}(x_1) = -1$.

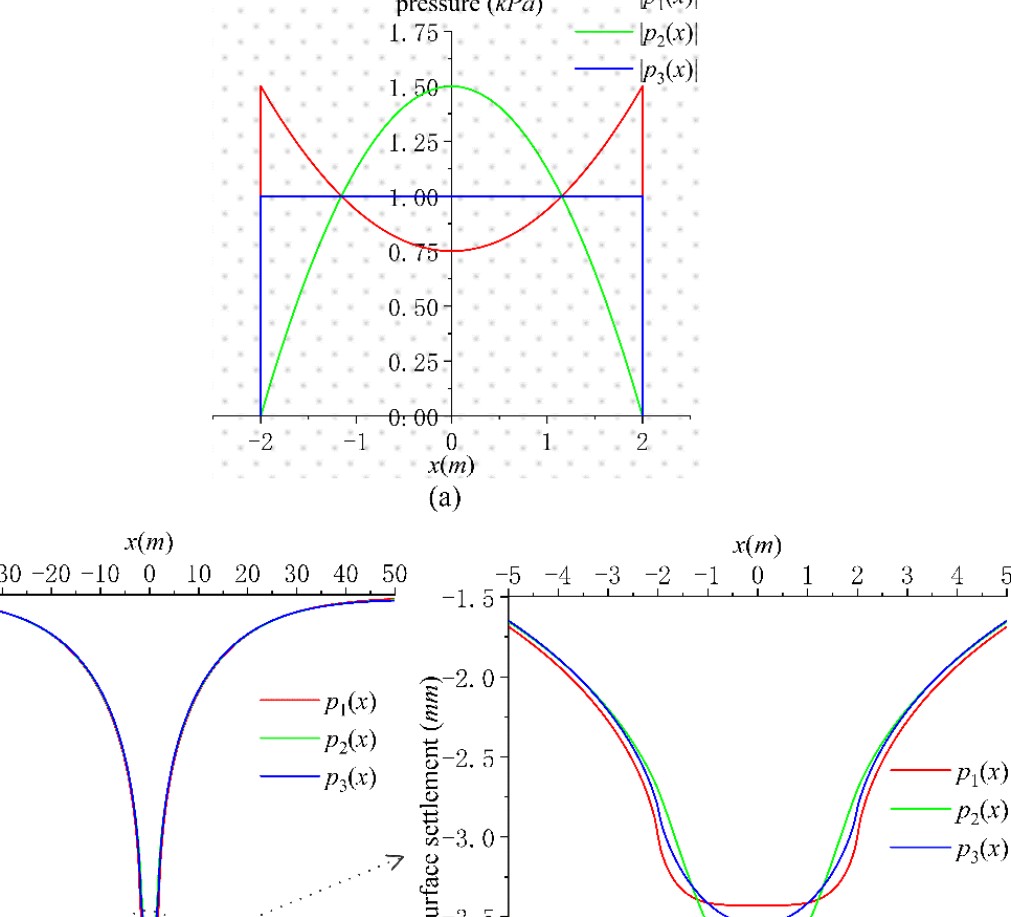

**Figure 10.** Three types of base pressure distribution and the surface settlement they caused: (**a**) distribution of base pressure, (**b**) surface settlement, and (**c**) surface settlement near foundation.

In order to compare the influence of load distribution on surface settlement, the resultant forces of the three types of base pressure distribution are the same, and they are all 4 kPa. The other parameters are the same as those described in Section 4.1, and the surface settlement that they caused are shown in Figure 10b,c.

It can be seen from Figure 10 that the surface settlement caused by the three types of base pressure distribution only shows certain differences near the foundation, and the maximum surface settlement occurs in the center of the foundation action area. When the load concentration at the center of the foundation is large, the surface settlement is the largest, followed by uniform distribution, and the surface settlement is smallest when the load concentration at the edge of the foundation is large. At the same time, it can be seen from Figure 10c that the surface settlement caused by the first distribution is approximately a horizontal line in the foundation action area. Only when the strip foundation is



absolutely rigid and it is subjected to axial compression, can the bottom of the foundation sink evenly. Therefore, we can infer that the distribution form of the base pressure under the absolutely rigid strip foundation is close to this distribution form.

### 4.4. Influence of Layered Soil on Surface Settlement and Additional Stress Distribution in Soil

When $H_1 = H_2 = 30 \, m$, $G_1 = 1.0 \, \text{MPa}$, $\mu_1 = \mu_2 = 0.2$, the distribution of base pressure is $p_y(x_1) = -3(4 - x_1^2)/8$, and other parameters are the same as those described in Section 4.1. Figure 11 shows the surface settlement curve when $G_2/G_1$ is 0.1, 0.5, 1.0, 2.0 and 10.0, respectively.

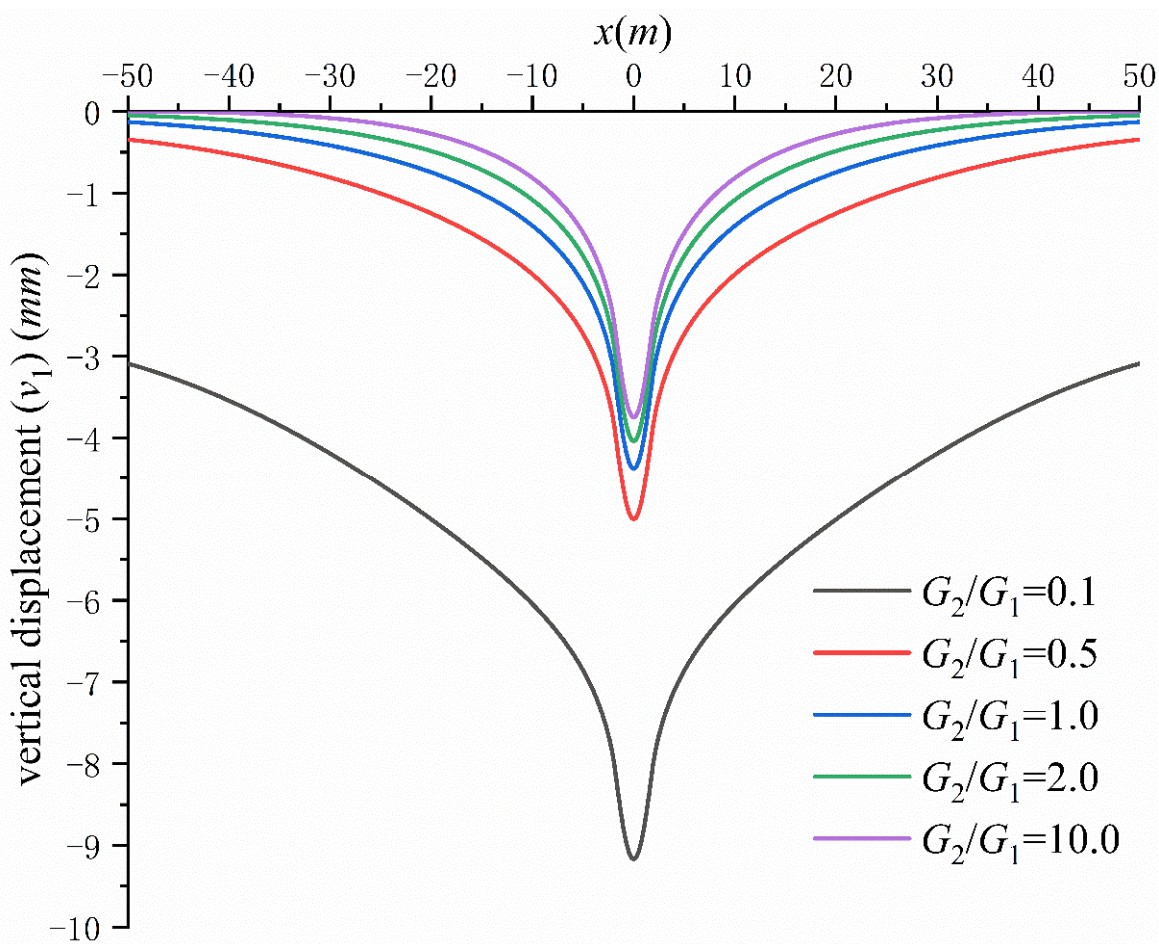

**Figure 11.** Effect of layered soil on surface settlement.

It can be seen from Figure 11 that the smaller the shear modulus ratio $G_2/G_1$, the greater the surface settlement. The influence of the change in $G_2/G_1$ on the surface settlement can be divided into two scenarios: when $G_2/G_1 > 1$, the change in $G_2/G_1$ has little influence on the surface settlement; when $G_2/G_1 < 1$, the change in $G_2/G_1$ has a great influence on the surface settlement.

Figure 12 shows the $\sigma_{1,x}$ curves (solid line) of the lower boundary of the first layer when $G_2/G_1$ is 0.1, 0.5, 1.0, 2.0 and 10.0 and the $\sigma_{2,x}$ curves (dotted line) of the upper boundary of the second layer when $G_2/G_1$ is 0.1, and 10.0. Figure 13 shows the $\sigma_{1,y}$ curve on the contact surface, and Figure 14 shows the $\tau_{1,xy}$ curve on the contact surface.

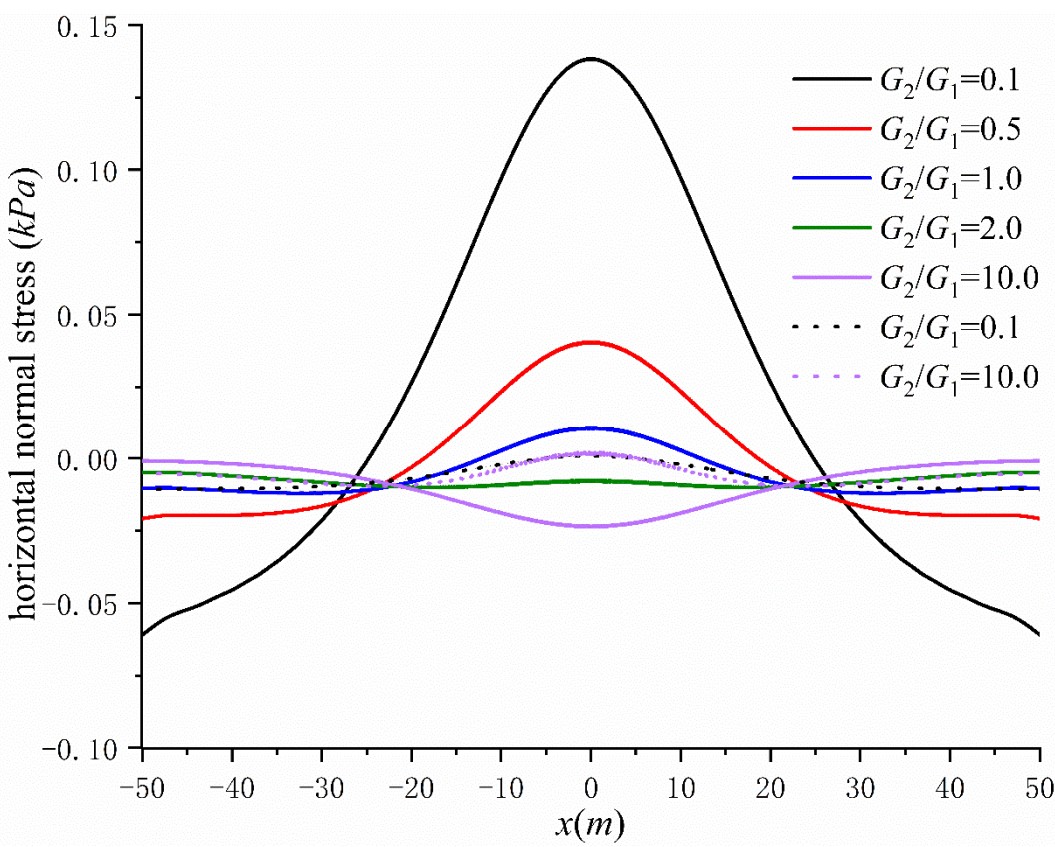

**Figure 12.** Effect of layered soil on horizontal normal stress at interface.

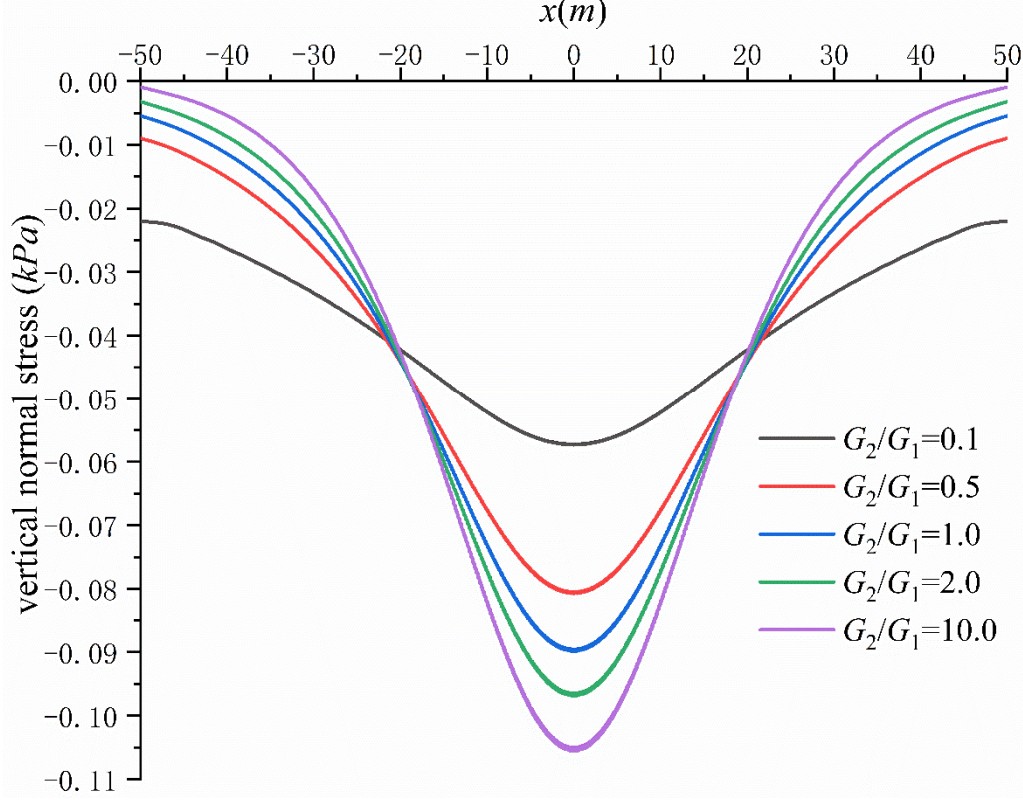

**Figure 13.** Effect of layered soil on vertical normal stress at interface.

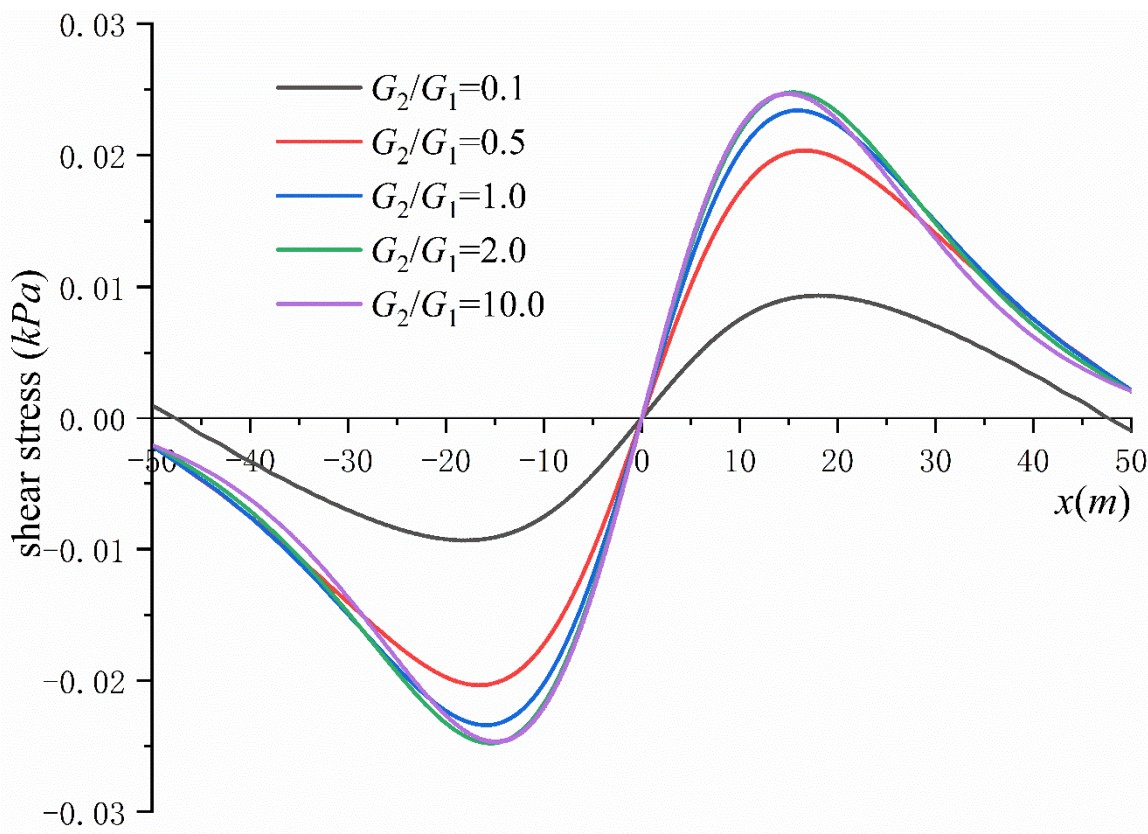

**Figure 14.** Effect of layered soil on shear stress at interface.

It can be seen from Figure 12 that with the decrease in the shear modulus ratio $G_2/G_1$ of the two layers of soil, the maximum value of $\sigma_{1,x}$ (absolute value) of the lower boundary of the first layer of soil decreases at first and then increases, and the tensile stress occurs. Moreover, with the decrease in $G_2/G_1$, the tensile stress increases. Compared with the first layer, $\sigma_{2,x}$ on the upper boundary of the second layer is very small and the change in $G_2/G_1$ has little effect on $\sigma_{2,x}$. Therefore, this paper only shows the $\sigma_{2,x}$ curves when $G_2/G_1$ is 0.1, 10.0 respectively.

As can be seen from Figure 13, when the soft soil layer is covered by the hard soil layer ($G_2/G_1 >$ 1), near $x = 0$, the $\sigma_{1,y}$ (absolute value) on the contact surface is larger than that of homogeneous soil ($G_2/G_1 = 1$), but in the distance, it is smaller than that of homogeneous soil, that is, the stress concentration phenomenon occurs near $x = 0$. When the hard soil layer covers the soft soil layer ($G_2/G_1 < 1$), the $\sigma_{1,y}$ on the contact surface is smaller near $x = 0$ than that of the homogeneous soil layer, but larger in the distance than that of the homogeneous soil layer, which is known as the stress diffusion phenomenon. At this time, the vertical normal stress distribution on the contact surface is relatively uniform, so the vertical deformation of the lower soil will be relatively uniform, but it can be seen from Figure 12 that the horizontal tensile stress of the lower boundary of the upper soil will be relatively large.

It can be seen from Figure 14 that with the decrease in $G_2/G_1$, the shear stress $\tau_{1,xy}$ (absolute value) on the contact surface gradually decreases. When $G_2/G_1 > 1$, the change in $G_2/G_1$ has little effect on the shear stress distribution on the contact surface, while when $G_2/G_1 < 1$, the change in $G_2/G_1$ has a greater effect on the shear stress.

Taking the same parameters as described in Section 4.1, since the base pressure is symmetrical about the $y$-axis, there are only two normal stress components on the $y$-axis, and the shear stress component is 0. Figure 15 shows the variation in the two normal stresses on the $y$-axis with depth $h$. At the surface, the horizontal normal stress is basically equal to the vertical normal stress. With the increase in $h$, the horizontal and vertical normal stress decay rapidly, and the decay speed of the horizontal normal stress is faster than that of the vertical normal stress. The horizontal normal stress at the surface is about $-1.5$, and it basically decays to zero at the depth of $h = 4\ m$. At the same time, it can be seen

that due to the different material properties of the two layers of soil, the horizontal normal stress curve jumps at $h = 30\ m$ (contact surface).

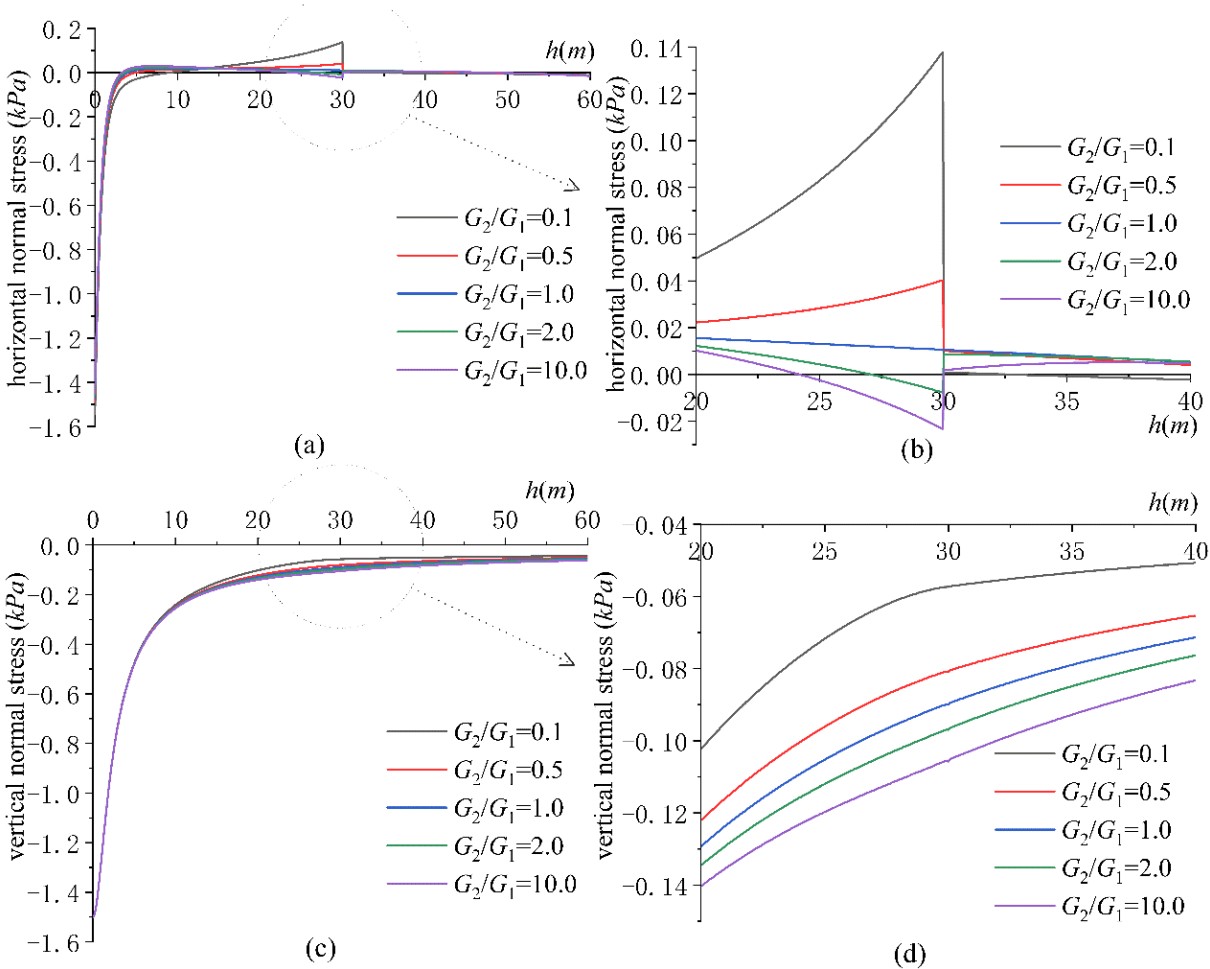

**Figure 15.** Distribution of horizontal normal stress and vertical normal stress along $y$-axis. (**a**) Horizontal normal stress, (**b**) Local diagram of horizontal normal stress, (**c**) Vertical normal stress and (**d**) Local diagram of vertical normal stress.

## 5. Conclusions

The method proposed in this paper can be used to calculate the stress and displacement of multi-layer soil above bedrock under the action of a strip foundation. In fact, it is an analytical method, and the accuracy of its calculation results is related to the number of terms taken by the analytic function. Through the programming calculation, it was found that when the number of terms of the analytic function is 550, the accuracy of the calculation results is quite high, which can well meet the stress boundary condition, displacement boundary condition, and the stress and displacement continuity conditions. The results are in good agreement with those obtained by the finite element method.

In this paper, the effects of different distribution forms of base pressure and layered soil on the surface settlement and stress distribution inside the soil were analyzed by numerical examples. The results show that the surface settlement curves corresponding to the three types of basement pressure distribution are basically the same, but there are some differences near the foundation action area. When the distribution of the base pressure reaches the maximum in the center of the base, the displacement is at its maximum. The displacement caused by uniform distribution takes second place. When the base pressure is largest on both sides of the base, the displacement is the smallest. In the foundation action area, the surface settlement curve is approximately a horizontal straight line, which proves that the base pressure of a vertical sinking rigid foundation is similar to this distribution form.

For two-layer soil, the surface settlement decreases with the increase in the ratio of shear modulus ($G_2/G_1$), $G_1$ and $G_2$ are the shear modulus of the upper and lower soil, respectively. When the $G_2/G_1$ increases:

(1)   The maximum horizontal normal stress of the lower boundary of the upper soil first decreases and then increases, and when $G_2/G_1$ reaches a certain value, the tensile stress begins to appear;
(2)   The maximum vertical normal stress on the contact surface increases, while the vertical normal stress decreases at a distance from the foundation;
(3)   The shear stress on the contact surface only changes significantly when $G_2 < G_1$.

**Author Contributions:** A.L. conceived and designed the study. X.S. derived the formulas and wrote the programs. X.S. wrote the paper. H.C. and C.Y. helped with drawing and translation. A.L., H.C. and C.Y. reviewed the edited manuscript. All authors have read and agreed to the published version of the manuscript.

**Funding:** The study is supported by the National Natural Science Foundation of China (Grant no.51974124).

**Institutional Review Board Statement:** Not applicable.

**Informed Consent Statement:** Not applicable.

**Data Availability Statement:** The data in the present study are available from the corresponding author upon reasonable request.

**Acknowledgments:** We thank the staff at the same laboratory.

**Conflicts of Interest:** The authors declare no conflict of interest.

## Appendix A

For $j = 1, \ldots, n, k = 1, \ldots, n_{j,1}$:

$$c_{j,ks} = \frac{\pi}{2H}k(1 - \sigma_{j,s}^2)\sigma_{j,s}^{k-1}, \ s = 1, \ldots, m_j + m_{j+1} + 2 \, ; \ k = 1, \ldots, n_{j,1} \tag{A1}$$

$$\begin{cases} d_{j,ks} = \frac{\pi}{4H}k(k-1)\left[\ln\left(\frac{\sin\theta_{j,s}}{1-\cos\theta_{j,s}}\right) - \frac{\pi}{2}i\right](1 - \sigma_{j,s}^2)^2\sigma_{j,s}^{k-2}, \ s = 1, \ldots, m_j + 1 \\ d_{j,ks} = \frac{\pi}{4H}k(k-1)\left[\ln\left(\frac{-\sin\theta_{j,s}}{1-\cos\theta_{j,s}}\right) + \frac{\pi}{2}i\right](1 - \sigma_{j,s}^2)^2\sigma_{j,s}^{k-2}, \ s = m_j + 2, \ldots, m_j + m_{j+1} + 2 \end{cases} \tag{A2}$$

$$\begin{cases} e_{j,ks} = \frac{\pi}{2H}k\left[\ln\left(\frac{\sin\theta_{j,s}}{1-\cos\theta_{j,s}}\right) - \frac{\pi}{2}i\right](1 - \sigma_{j,s}^2)\sigma_{j,s}^k, \ s = 1, \ldots, m_j + 1 \\ e_{j,ks} = \frac{\pi}{2H}k\left[\ln\left(\frac{-\sin\theta_{j,s}}{1-\cos\theta_{j,s}}\right) + \frac{\pi}{2}i\right](1 - \sigma_{j,s}^2)\sigma_{j,s}^k, \ s = m_j + 2, \ldots, m_j + m_{j+1} + 2 \end{cases} \tag{A3}$$

For $j = 1, \ldots, n, k = 1, \ldots, n_{j2}$:

$$f_{j,ks} = \frac{\pi}{2H}k(1 - \sigma_{j,s}^2)\sigma_{j,s}^{k-1}, \ s = 1, \ldots, m_j + m_{j+1} + 2 \tag{A4}$$

For $j = 1, \ldots, n, k = 1, \ldots, n_{j,1}$:

$$g_{j,ks} = \frac{1}{G_j}\kappa_j\sigma_{j,s}^k, \ s = 1, \ldots, m_j + m_{j+1} + 2 \tag{A5}$$

$$\begin{cases} h_{j,ks} = \frac{1}{G_j}\frac{k}{2}\left[\ln\left(\frac{\sin\theta_{j,s}}{1-\cos\theta_{j,s}}\right) + \frac{\pi}{2}i\right](\sigma_{j,s}^2 - 1)\sigma_{j,s}^{-1-k}, \ s = 1, \ldots, m_j + 1 \\ h_{j,ks} = \frac{1}{G_j}\frac{k}{2}\left[\ln\left(\frac{-\sin\theta_{j,s}}{1-\cos\theta_{j,s}}\right) - \frac{\pi}{2}i\right](\sigma_{j,s}^2 - 1)\sigma_{j,s}^{-1-k}, \ s = m_j + 2, \ldots, m_j + m_{j+1} + 2 \end{cases} \tag{A6}$$

$$l_{j,ks} = \frac{1}{G_j}\sigma_{j,s}^{-k}, \ s = 1, \ldots, m_j + m_{j+1} + 2; \ k = 1, \ldots, n_{j2} \tag{A7}$$

$$
f_s = \begin{cases} 0, & s = 1, \ldots, m_{1,1} \\ p_y(x_{1,s}) + i p_x(x_{1,s}), & s = m_{1,1}+1, \ldots, m_{1,1}+m_{1,2}+1 \\ 0, & s = m_{1,1}+m_{1,2}+2, \ldots, m_{1,1}+m_{1,2}+m_{1,3}+1 \end{cases} \tag{A8}
$$
$$
x_{1,s} = \frac{H}{\pi} \ln\left(\frac{\sin\theta_{1,s}}{1-\cos\theta_{1,s}}\right)
$$

### Appendix B

For $j = 1, \ldots, n$:

$$
t_{j,1} = \omega_j(\zeta_j) \tag{A9}
$$

For the upper boundary of the $j$-th layer:

$$
t_{j,1} = \omega_j(\sigma_j) = \frac{H_j}{\pi}\left[\ln\left(\frac{\sin\theta_j}{1-\cos\theta_j}\right) + \frac{\pi}{2}i\right], \ \theta_j \in (0,\pi) \tag{A10}
$$

For the lower boundary of the $j$-th layer:

$$
t_{j,1} = \omega_j(\sigma_j) = \frac{H_j}{\pi}\left[\ln\left(\frac{-\sin\theta_j}{1-\cos\theta_j}\right) - \frac{\pi}{2}i\right], \ \theta_j \in (\pi, 2\pi) \tag{A11}
$$

$$
t_{j,2} = \omega'_j(\zeta_j) = \frac{2H_j}{\pi(1-\zeta_j^2)} \tag{A12}
$$

$$
t_{j,3} = \omega''_j(\zeta_j) = \frac{4H_j\zeta_j}{\pi(1-\zeta_j^2)^2} \tag{A13}
$$

$$
t_{j,4} = \varphi_j(\zeta_j) = \sum_{k=0}^{n_{j,1}} a_{j,k}\zeta_j^k \tag{A14}
$$

$$
t_{j,5} = \varphi'_j(\zeta_j) = \sum_{k=1}^{n_{j,1}} k a_{j,k}\zeta_j^{k-1} \tag{A15}
$$

$$
t_{j,6} = \varphi''_j(\zeta_j) = \sum_{k=2}^{n_{j,1}} k(k-1) a_{j,k}\zeta_j^{k-2} \tag{A16}
$$

$$
t_{j,7} = \psi_j(\zeta_j) = \sum_{k=1}^{n_{j2}} b_k\zeta_j^k \tag{A17}
$$

$$
t_{j,8} = \psi'_j(\zeta_j) = \sum_{k=1}^{n_{j2}} k b_{j,k}\zeta_j^{k-1} \tag{A18}
$$

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
