# Peer review of "Complex Variable Solution for Stress and Displacement of Layered Soil with Finite Thickness"

_applsci, doi:10.3390/app12020766_

Round 1

Reviewer 1 Report

The Manuscript ID applsci-1543898 titled "Complex Variable Solution for Stress and Displacement of Layered Soil with Finite Thickness" are recommended to be accepted after minor corrections. The paper has good organisation and good flow. The authors have good hold in the subject area. The analytical formulations are well organised and expressed. The paper is well discussed and conclusion is good.   However, there are a few observations: 1. The paper requires minor English language editing and some proofreading. 2. The introduction should be improved with more references. 3. The ANSYS model should be described better and the boundary conditions for the model should be stated. 4. The authors should show an image of the ANSYS model used. 5. The theory document or reference document for the ANSYS platform should be cited and reference added. Also, add the version of the ANSYS model. 6. Construct the abstract from 3 paragraphs to 1 praragraph, and keep it into 200 words. 7. Revise the Section 1-Introduction into 3 paragraphs as the authors have 5 paragraphs.

Reviewer 2 Report

see my comments in the attached file
